# Phosphorylation of WDR48 by phototropins drives starch degradation to promote stomatal opening

Shota Yamauchi[1,2], Saashia Fuji[1], Hiroki Ikuta[1], Naoyuki Sugiyama[3,4], Yutaka Kodama [5], Luca Distefano[6], Haruki Fujii[7], Kota Yamashita [8], Hinano Takase[8], Mika Nomoto [9,10], Yasuomi Tada [9,10], Taishi Umezawa [8], Kazuhiro Hotta [11], Diana Santelia [6], Ken-ichiro Shimazaki[12] & Atsushi Takemiya [1] ✉

Plant stomata open in response to blue light under a background of red light. Red light induces starch accumulation through guard cell photosynthesis, whereas blue light causes rapid starch degradation via the activation of plasma membrane H⁺-ATPase through phosphorylation of BLUS1 by phototropins, thus promoting stomatal opening. However, how phototropins mediate starch degradation remains unclear. In this study, we identified WD-repeat protein 48 (WDR48), which is essential for starch degradation. Phosphoproteomic analysis revealed that WDR48 was phosphorylated in response to blue light. Mutation of *WDR48* impaired starch degradation and stomatal opening by blue light. Phototropins interacted with and phosphorylated WDR48. WDR48 and BLUS1 constituted separate signalling pathways required for starch degradation. We propose that the coordinated control of starch synthesis by red light and its degradation by blue light signalling through WDR48 and BLUS1 is a key mechanism for stomatal opening by red and blue light.

Plant stomata, which are formed by a pair of guard cells, are critical in regulating gas exchange between plants and the atmosphere. These specialised cells adjust stomatal pore size in response to changing environmental conditions[1–3]. Light triggers stomatal opening, facilitating the uptake of carbon dioxide ($CO_2$) for photosynthesis as well as the transport of water and nutrients from the roots via transpiration. Stomatal responses to light can be divided into photosynthesis-dependent and blue light-dependent responses[2,4]. The photosynthesis-dependent response, which can be activated by red light, is induced by continuous high fluence rates and eliminated by 3-(3,4-dichlorophenyl)−1,1-dimethylurea (DCMU), a photosynthetic electron transport inhibitor[5,6]. Guard cells and mesophyll cells contain photosynthetically active chloroplasts, contributing to red light-induced stomatal opening[7–9]. In contrast, the blue light response occurs at low fluence rates and relies on photoreceptor phototropin-coupled ion transport and metabolic rearrangements in the guard cells[10,11].

[1]Department of Biology, Graduate School of Sciences and Technology for Innovation, Yamaguchi University, Yamaguchi, Japan. [2]Department of Applied Biological Science, Faculty of Science and Technology, Tokyo University of Science, Noda, Chiba, Japan. [3]Department of Molecular & Cellular BioAnalysis, Graduate School of Pharmaceutical Sciences, Kyoto University, Kyoto, Japan. [4]Omics Research Center, National Cerebral and Cardiovascular Center, Suita, Osaka, Japan. [5]Center for Bioscience Research and Education, Utsunomiya University, Tochigi, Japan. [6]Institute of Integrative Biology, ETH Zürich, Zürich, Switzerland. [7]Department of Electrical and Electronic Engineering, Graduate School of Science and Technology, Meijo University, Nagoya, Japan. [8]Graduate School of Bio-Applications and Systems Engineering, Tokyo University of Agriculture and Technology, Tokyo, Japan. [9]Center for Gene Research, Nagoya University, Nagoya, Japan. [10]Division of Biological Science, Graduate School of Science, Nagoya University, Nagoya, Japan. [11]Department of Electrical and Electronic Engineering, Faculty of Science and Technology, Meijo University, Nagoya, Japan. [12]Department of Biology, Faculty of Science, Kyushu University, Fukuoka, Japan. ✉e-mail: take.pcs@yamaguchi-u.ac.jp

Furthermore, an interaction occurs between photosynthesis- and blue light-dependent responses. The blue light response is enhanced when blue light is superimposed on a strong red light background and simultaneous irradiation with blue and red light results in greater stomatal opening than irradiation with blue or red light alone[2,12–16]. The synergistic effect of blue and red light on stomatal opening may be important for guard cells to integrate information from blue light and photosynthesis to appropriately control stomatal opening and $CO_2$ uptake under photosynthetically active conditions. Studies have suggested that both blue light signals and red light-driven mesophyll cell photosynthesis activate the plasma membrane $H^+$-ATPase and inactivate S-type anion channels, thereby contributing synergistically to stomatal opening in intact leaves[17–19]. This synergistic effect of blue and red light on stomatal opening has also been observed in isolated epidermis, suggesting the presence of mesophyll cell-independent regulation[15]. However, the underlying mechanisms remain elusive.

Phototropins are plant-specific blue light receptor kinases containing two light, oxygen, or voltage domains (LOV1 and LOV2) at the N-terminus, each of which binds a flavin mononucleotide (FMN) as a chromophore[20]. Light perception, mainly through the LOV2 domain, activates the C-terminal kinase domain and induces receptor autophosphorylation[21]. Light-activated phototropins phosphorylate their substrate BLUE LIGHT SIGNALING1 (BLUS1), a guard cell-specific Ser/Thr protein kinase[22,23]. BLUS1 contains a C-terminal regulatory domain that represses the N-terminal kinase domain. Phototropins phosphorylate a conserved Ser residue within the regulatory domain of BLUS1, alleviating suppression of the kinase domain[16]. The activated BLUS1 transduces signals via downstream elements such as a Raf-like MAP kinase kinase kinase (MAPKKK) BLUE LIGHT-DEPENDENT $H^+$-ATPASE PHOSPHORYLATION (BHP) and type 1 protein phosphatase (PP1), ultimately activating the plasma membrane $H^+$-ATPase via the phosphorylation of two Thr residues within the C-terminal autoinhibitory domain[17,18,24–26]. In addition to BLUS1, phototropins phosphorylate a Raf-like MAPKKK CONVERGENCE OF BLUE LIGHT AND $CO_2$ 1 (CBC1)[27]. CBC1 and its paralogue CBC2 inhibit plasma membrane S-type anion channels in response to blue light[27,28]. Consequent $H^+$-ATPase activation and anion channel inhibition drive plasma membrane hyperpolarisation, providing the driving force for $K^+$ influx through voltage-gated inward $K^+$ channels[29]. $K^+$ accumulation drives water uptake, causing swelling of guard cells and stomatal opening.

Simultaneously with the regulation of membrane potential, phototropins induce rapid starch degradation in guard cell chloroplasts[30]. Chloroplasts are a predominant feature of guard cells as they are not found in other leaf epidermal cells. Guard cells partially synthesise starch via autonomous photosynthetic activity[31,32]. The degradation products of starch by blue light are converted to glucose, contributing to maintaining cytosolic sugar homeostasis needed for stomatal opening[30,33]. Starch degradation products have also been suggested to serve as a precursor of malate, which acts as a counter-ion to $K^+$ influx[2,34]. Mutation of BLUS1 or AUTOINHIBITED $H^+$-ATPASE 1 (AHA1), a major isoform of plasma membrane $H^+$-ATPase in guard cells, impairs starch degradation by blue light, revealing that BLUS1-mediated $H^+$-ATPase activation is involved in this process[30,35]. Similarly, mutation of the starch degradation enzymes β-AMYLASE 1 (BAM1) and α-AMYLASE 3 (AMY3) impairs blue light-induced starch degradation and stomatal opening[30]. However, the mechanisms by which phototropins cause rapid starch degradation in guard cell chloroplasts remain unclear.

In this study, we performed phosphoproteomic analysis to uncover novel factors involved in phototropin-controlled stomatal opening. We identified a novel phototropin kinase substrate, WD-repeat protein 48 (WDR48), that is essential for starch degradation and stomatal opening by blue light. WDR48 and BLUS1 form separate signalling pathways implicated in blue light-dependent starch degradation in guard cells.

## Results

### WDR48 is phosphorylated in response to blue light and regulates stomatal opening

We conducted a phosphoproteomic analysis using Arabidopsis thaliana guard cell protoplasts from the wild type and the phot1-5 phot2-1 mutant, and analysed proteins whose phosphorylation levels changed in response to blue light in the wild type but not in the phot1-5 phot2-1 mutant[17,22,27]. We identified a WD-repeat protein (WDR), At3g05090, known as LATERAL ROOT STIMULATOR 1 (LRS1)[36], which was phosphorylated in a blue light- and phototropin-dependent manner (Fig. 1a, b). This WDR protein is conserved amongst eukaryotes and is generally referred to as WDR48[37]. We, therefore, refer to this protein as WDR48 in this study. WDR48 contains six WD40 repeats at the N-terminus and a RAWUL domain at the C-terminus that appears to function in protein–protein interactions[37]. Ser-393, which is located between the two domains, was identified as a blue light-dependent phosphorylation site (Fig. 1c).

We generated phospho-specific antibodies against Ser-393 of WDR48 and found that Ser-393 was weakly phosphorylated under red light, whereas phosphorylation was enhanced upon blue light irradiation in the wild type and absent in the phot1-5 phot2-1 mutant (Fig. 1d, e). The bands observed under both red and blue light conditions disappeared after λ-protein phosphatase (λ-PPase) treatment in the wild type (Supplementary Fig. 1a, b) and were absent in both the wdr48-1 mutant and the phosphodefective S393A mutant described later (Supplementary Fig. 1c–f), suggesting that the phospho-specific antibodies generated in this study specifically recognise phosphorylated WDR48.

To determine the physiological role of WDR48 in the stomatal opening, we obtained two T-DNA insertion mutants, wdr48-1 (GABI_585_E01) and wdr48-2 (SALK_059570C), harbouring a T-DNA insertion in the $2^{nd}$ and $14^{th}$ introns, respectively (Fig. 1f). Immunoblot analysis using anti-WDR48 antibodies revealed that both wdr48 mutants were null alleles (Fig. 1g). We confirmed that there were no significant differences in stomatal size or density between the wild-type and wdr48 mutants (Supplementary Fig. 2). We then examined the stomatal opening in the epidermal peels, and wdr48-1 and wdr48-2 plants exhibited defects in the stomatal opening in response to blue light, similar to the phot1-5 phot2-1 mutant (Fig. 1h).

To examine whether the wdr48 mutant impairs the activation of plasma membrane $H^+$-ATPase by blue light, we measured $H^+$ pumping in guard cell protoplasts. The wdr48-1 mutant exhibited the same level of blue light-dependent $H^+$ pumping activity as the wild-type (Fig. 1i, j). The phosphorylation of $H^+$-ATPase in response to blue light was not affected in the wdr48-1 mutant (Fig. 1k, l). The electrophoretic mobility shift on SDS-PAGE confirmed the blue light-dependent autophosphorylation of phototropins in the wdr48-1 mutant (Fig. 1m). Furthermore, blue light-dependent phosphorylation of BLUS1 was prominent in the wdr48-1 mutant (Fig. 1n). Overall, these results suggest that WDR48 stimulates stomatal opening by regulating mechanisms that are distinct from $H^+$-ATPase activation.

### WDR48 regulates stomatal opening via the regulation of starch dynamics in guard cells

To determine whether wdr48 impairs blue light-dependent starch degradation, we quantified the starch in guard cells using the pseudo-Schiff propidium iodide (PS-PI) staining technique[30,38]. We prepared leaf epidermis from dark-adapted wild-type plants and observed little starch accumulation in guard cells in the dark (Fig. 2a, b). Irradiation of the epidermis with red light induced starch accumulation, and subsequent exposure to blue light caused starch degradation. The application of DCMU to the epidermis suppressed starch accumulation by red light and starch degradation by blue light (Fig. 2a, b). The wdr48-1 mutant showed starch accumulation in guard cells under red light but

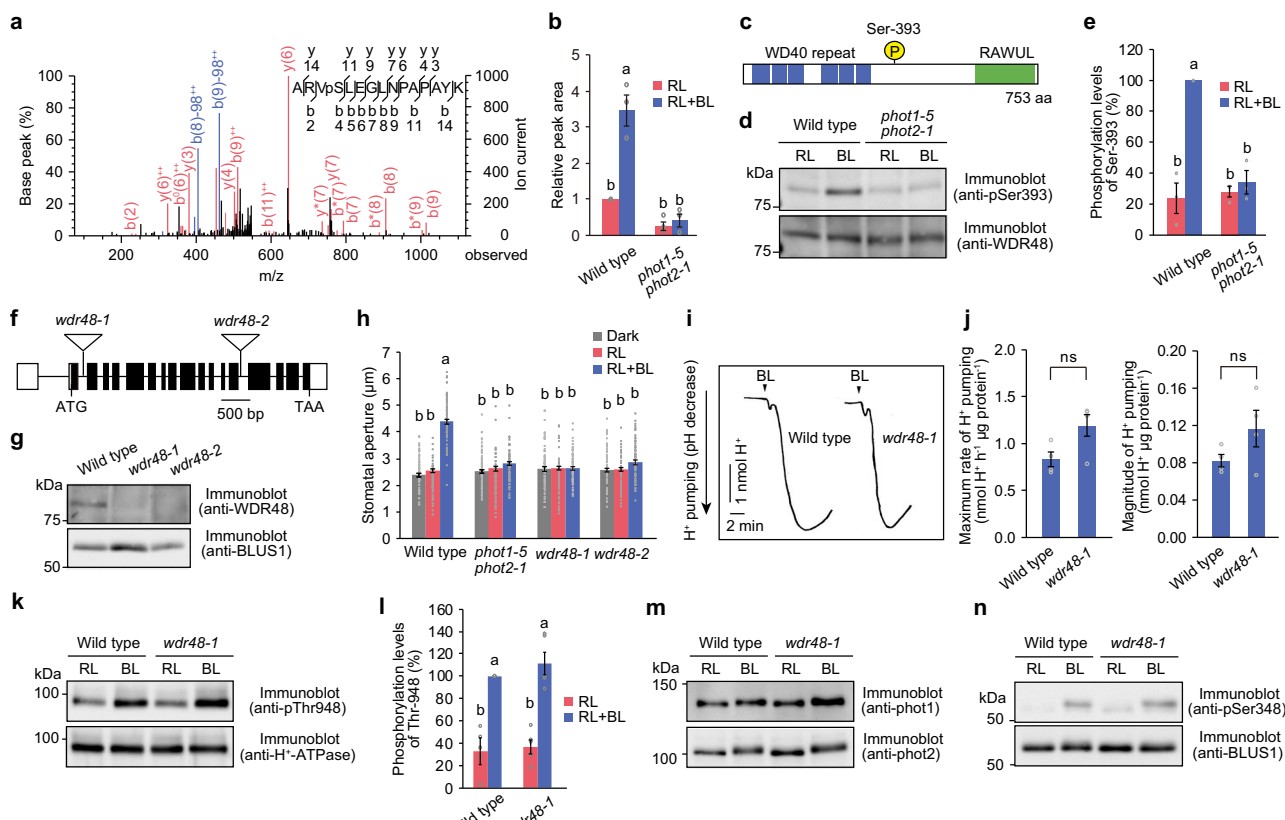

**Fig. 1 | Blue light-dependent phosphorylation of WDR48 and stomatal phenotype in the *wdr48* mutants.** Phosphoproteomic analysis. MS/MS spectrum (**a**) and relative peak area (**b**) of the phosphopeptide containing pSer-393 of WDR48. Guard cell protoplasts prepared from the wild type and *phot1 phot2* mutant were illuminated by red light (RL: 600 µmol m⁻² s⁻¹) for 30 min, after which a pulse of blue light (BL: 100 µmol m⁻² s⁻¹, 30 s) was superimposed on the background of RL. The labels "−98", "*", and "°" represent the neutral loss of H₃PO₄, NH₃, and H₂O, respectively, from b and y ions. **c** Domain structure of WDR48. **d**, **e** Phosphorylation of WDR48 at Ser-393. Guard cell protoplasts were illuminated by RL (300 µmol m⁻² s⁻¹) for 30 min, after which a pulse of BL (100 µmol m⁻² s⁻¹, 30 s) was superimposed on the RL. **f**, Gene structure of *WDR48* and sites of T-DNA insertion in the *wdr48* mutants. Bar indicates 500 bp. **g**, Expression of WDR48 protein in guard cell protoplasts. **h**, Light-dependent stomatal opening. Epidermal peels were illuminated with RL (50 µmol m⁻² s⁻¹) or RL and BL (10 µmol m⁻² s⁻¹) for 2 h. **i**, **j**, H⁺ pumping. Guard cell protoplasts were pre-illuminated with RL (300 µmol m⁻² s⁻¹) for 2 h, after which a pulse of BL (100 µmol m⁻² s⁻¹, 30 s) was superimposed on the RL. **k**, **l**, Phosphorylation of H⁺-ATPase. **m**, Autophosphorylation of phot1 and phot2. The autophosphorylation was detected as an electrophoretic mobility shift by immunoblotting. **n**, Phosphorylation of BLUS1 at Ser-348. For (**e**) and (**e**), data represent mean ± SEM (*n* = 3 biologically independent samples; One-way ANOVA with Tukey's test, *P* < 0.01). For (**h**), data represent mean ± SEM (*n* = 75 stomata examined over three independent experiments; One-way ANOVA with Tukey's test, *P* < 0.01). For (**j**), data represent mean ± SEM (*n* = 4 biologically independent samples). ns indicates no significant difference (Student's *t*-test, *P* < 0.05). For (**l**), data represent mean ± SEM (*n* = 5 biologically independent samples; One-way ANOVA with Tukey's test, *P* < 0.01). For (**g**), (**m**), and (**n**), similar results were obtained in three independent experiments.

---

failed to cause starch degradation by blue light, similar to the *phot1-5 phot2-1* mutant (Fig. 2c, d).

We determined the time course of starch degradation and stomatal opening in response to blue light. When wild-type epidermis was irradiated with blue light following red light treatment to ensure starch accumulation, guard cell starch was rapidly degraded within 5 min (Fig. 2e, f). Consistent with starch degradation, the stomatal opening was prominent 5 min after blue light irradiation and peaked at approximately 10 min (Fig. 2g). In contrast, no starch degradation or stomatal opening was detected in *phot1-5 phot2-1* and *wdr48-1* mutants after 30 min of blue light irradiation (Fig. 2e–g). These findings suggest that WDR48 controls phototropin-mediated starch degradation to allow rapid stomatal opening in response to blue light.

Phototropins also regulate chloroplast movements in addition to stomatal opening[20]. Because WDR48 controls starch degradation in chloroplasts, we examined whether chloroplast movements are impaired in the *wdr48* mutants using the green and white band assays[39]. The *wdr48* mutants exhibited chloroplast accumulation under weak blue light and avoidance under strong blue light, similar to

the wild type (Supplementary Fig. 3), suggesting that WDR48 is unlikely to function in chloroplast movements.

## Phosphorylation of WDR48 is necessary for starch degradation and stomatal opening

To examine the physiological role of blue light-dependent phosphorylation of WDR48 in stomatal responses, we generated transgenic plants expressing phosphodefective or phosphomimetic forms of WDR48, in which Ser-393 was replaced with Ala (S393A) or Glu (S393E), respectively, as an N-terminal fusion to GFP in the *wdr48-1* mutant background under the control of its promoter. We selected transgenic lines that showed comparable levels of expression to endogenous WDR48 by immunoblot analysis using anti-WDR48 antibodies (Supplementary Fig. 4). Expression of GFP-WDR48 and GFP-S393E complemented the inhibition of starch degradation and stomatal opening in response to blue light in the *wdr48-1* mutant (Fig. 3a–c). In contrast, when GFP-S393A was introduced into the *wdr48-1* mutant, neither starch degradation nor stomatal opening was restored.

We evaluated the stomatal opening in intact leaves using a gas exchange system. Strong red light irradiation increased stomatal

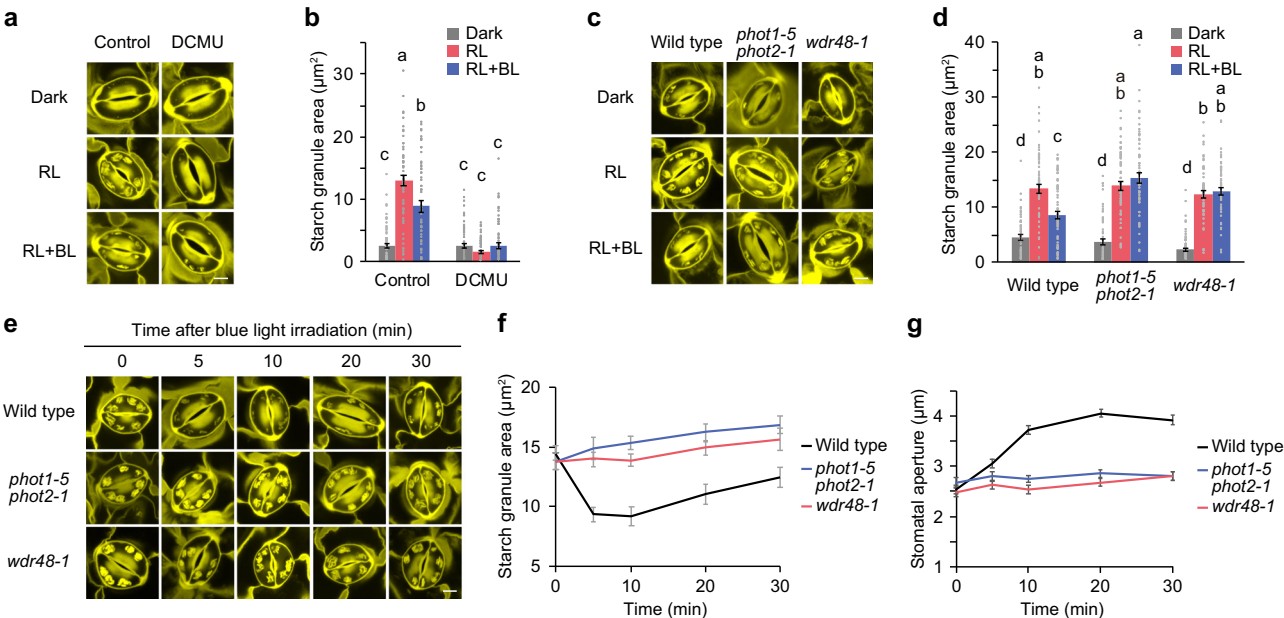

**Fig. 2 | Impairments of blue light-dependent starch degradation in the *wdr48* mutant. a, b** Light-induced changes in guard cell starch amount. Epidermal peels from dark-adapted leaves were pre-illuminated with red light (RL: 50 μmol m$^{-2}$ s$^{-1}$) for 2 h in the presence or absence of 10 μM DCMU, after which blue light (BL: 10 μmol m$^{-2}$ s$^{-1}$) was superimposed on the background of red light for 10 min. Confocal images of PS-PI staining of the starch granules (**a**). Quantification of starch granule area (**b**). **c, d** Light-induced changes in guard cell starch in the *wdr48* mutant. Epidermal stripes from dark-adapted leaves were pre-illuminated with RL for 2 h, after which BL was superimposed on the background of RL for 30 min. Confocal images of PS-PI staining of the starch granules (**c**). Quantification of starch granule area (**d**). **e–g** Time course of BL-induced starch degradation and stomatal opening. Epidermal stripes from dark-adapted leaves were pre-illuminated with RL for 2 h, after which BL was superimposed on the background of RL for the indicated time periods. Confocal images of PS-PI staining of starch granules (**e**). Quantification of starch granule area (**f**). Measurement of stomatal aperture (**g**). For (**a, c, e**), each bar represents 5 μm. For (**b, d**), data represent mean ± SEM (*n* = 60 stomata examined over three independent experiments; One-way ANOVA with Tukey's test, *P* < 0.05). For (**f**), data represent mean ± SEM (*n* = 60 stomata from three independent experiments). For (**g**), data represent mean ± SEM (*n* = 75 stomata examined over three independent experiments.

conductance as a result of photosynthesis-dependent stomatal opening, and the superimposition of weak blue light further promoted stomatal opening (Fig. 3d). When compared to the wild-type plants, both red light- and blue light-dependent stomatal opening were diminished in the *wdr48-1* mutant (Fig. 3e, f). Furthermore, GFP-WDR48 and GFP-S393E lines exhibited enhanced stomatal conductance in response to red and blue light. In contrast, the phosphodefective GFP-S393A line complemented the *wdr48-1* stomatal phenotype under red light but failed to restore blue light-dependent stomatal opening. These findings suggest that phosphorylation of WDR48 is crucial for starch degradation and stomatal opening in response to blue light, and that WDR48 also plays a role in regulating stomatal opening under red light via a phosphorylation-independent mechanism.

In addition to its role in stomatal opening, WDR48 has been shown to be implicated in lateral root formation[36]. In the *wdr48* mutants, both lateral root formation and primary root growth were impaired (Supplementary Fig. 5). These defects were rescued not only by expression of wild-type WDR48 but also by phosphodefective and phosphomimetic forms of WDR48, suggesting that WDR48 regulates root development through a function that is independent of its phosphorylation.

### WDR48 is a direct substrate of phototropin kinases

We further explored the mechanism of WDR48 phosphorylation by phototropins. Blue light-dependent phosphorylation of WDR48 was detected in guard cell protoplasts in the wild type but not in the *phot1-5 phot2-1* double mutant (Fig. 4a, b). WDR48 phosphorylation was also detected in the *phot1-5* and *phot2-1* single mutants, indicating that phot1 and phot2 mediate WDR48 phosphorylation. Consistent with this finding, starch granules were degraded in the *phot1-5* and *phot2-1* mutants but not in the *phot1-5 phot2-1* mutant (Supplementary Fig. 6a, b). We evaluated the phosphorylation of WDR48 in mutants of BLUS1

and CBC kinases, which are phosphorylation substrates of phototropins[22,27]. WDR48 phosphorylation by blue light was prominent in the *blus1-1* and *cbc1-1 cbc2-1* mutants (Fig. 4c, d), indicating that WDR48 phosphorylation is mediated by blue light signalling separate from the BLUS1 and CBC kinases.

We then analysed the phosphorylation time course of WDR48 in greater detail. WDR48 was phosphorylated 15 s after blue light pulse irradiation, peaked at 2 min, and was dephosphorylated within 20 min (Fig. 4e, f). The phosphorylation time course of WDR48 was consistent with that of BLUS1 (Fig. 4e, f). Therefore, we hypothesised that WDR48 is a phosphorylation substrate of phototropins, such as BLUS1 and CBC1. To test this possibility, we produced recombinant FLAG-WDR48 and His-phot1 or His-phot2 proteins using an in vitro transcription/translation system and performed an in vitro kinase assay. Phosphorylation of WDR48 at Ser-393 was detected when FLAG-WDR48 was incubated with His-phot1 or His-phot2 under blue light, but not in darkness (Fig. 4g, h and Supplementary Fig. 7a, b). Blue light-dependent phosphorylation of FLAG-WDR48 was not observed when incubated with the kinase-dead forms His-phot1 (D806N) or His-phot2 (D720N). Furthermore, λ-PPase treatment abolished the blue light-dependent phosphorylation of FLAG-WDR48 by His-phot1 or His-phot2 (Supplementary Fig. 7c–f).

The in vivo interaction between WDR48 and phot1 was confirmed by a co-immunoprecipitation assay using transgenic plants expressing 3×FLAG-WDR48. Endogenous phot1 protein was co-precipitated with 3×FLAG-WDR48 irrespective of blue light irradiation (Fig. 4i, j). We further verified direct binding between FLAG-WDR48 and HA-phot1 by an in vitro pull-down assay, independent of blue light (Fig. 4k, l). To identify the region of phot1 responsible for the interaction, we performed an in vitro pull-down assay with FLAG-phot1 fragments. His-WDR48 bound to the C-terminal kinase domain of phot1 (FLAG-P1C)

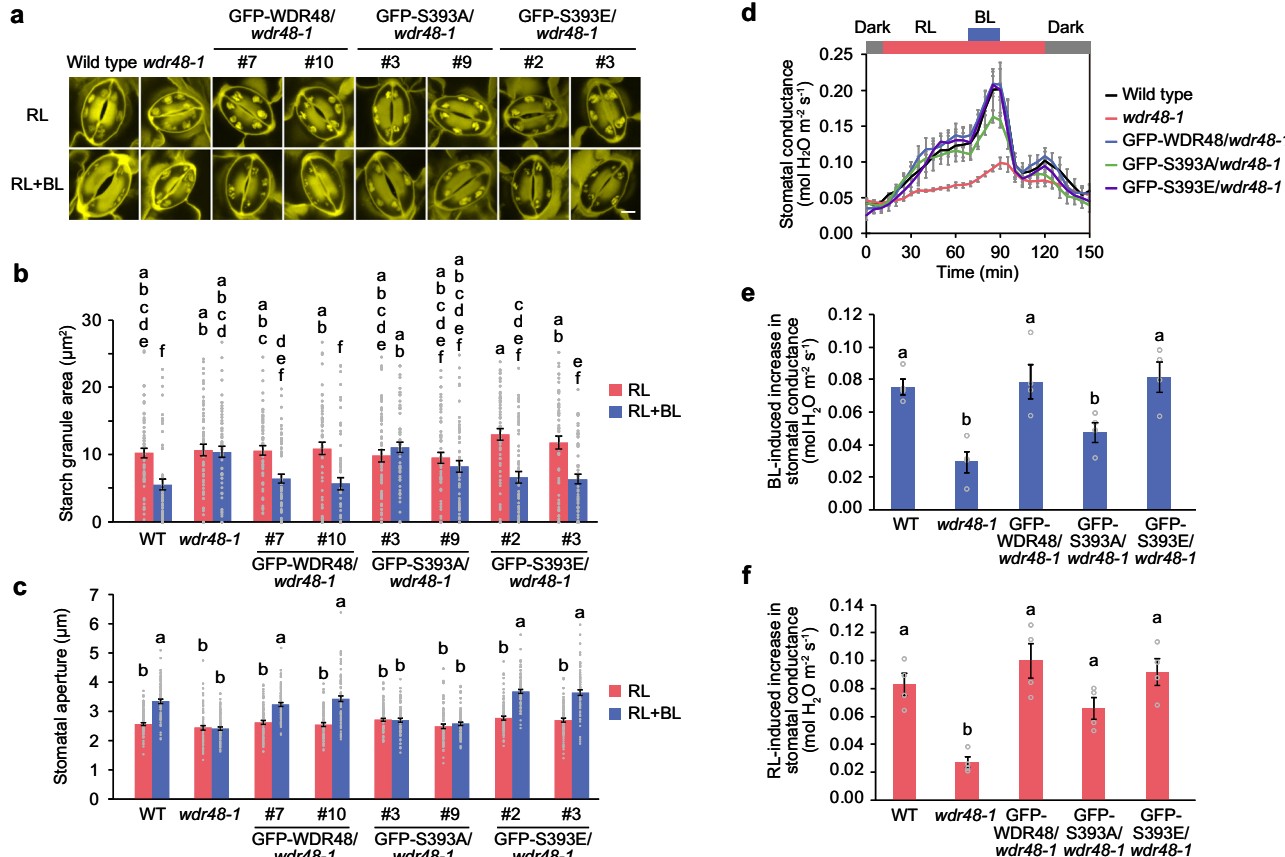

**Fig. 3 | Phosphorylation of WDR48 is essential for blue light-induced starch degradation and stomatal opening. a–c** Light-induced changes in starch dynamics and stomatal opening in transgenic plants expressing phosphodefective and phosphomimetic WDR48 variants. Epidermal stripes from dark-adapted leaves were pre-illuminated with red light (RL: 50 μmol m$^{-2}$ s$^{-1}$) for 2 h, after which blue light (BL: 10 μmol m$^{-2}$ s$^{-1}$) was superimposed on the background of RL for 10 min. Confocal images of PS-PI staining of the starch granules (**a**). Bar represents 5 μm. Quantification of the starch granule area (**b**). Data represent mean ± SEM ($n = 60$ stomata examined over three independent experiments; One-way ANOVA with Tukey's test, $P < 0.05$. Measurement of the stomatal aperture (**c**). Data represent mean ± SEM ($n = 75$ stomata examined over three independent experiments; One-way ANOVA with Tukey's test, $P < 0.01$). **d–f** Light-dependent changes in stomatal conductance in intact leaves. Leaves of dark-adapted plants were illuminated with RL (300 μmol m$^{-2}$ s$^{-1}$) for 1 h, after which BL (10 μmol m$^{-2}$ s$^{-1}$) was superimposed on the background of RL for 20 min (**d**). Quantification of stomatal conductance changes in response to BL (**e**) and RL (**f**). Data represent mean ± SEM ($n = 4$ biologically independent samples; One-way ANOVA with Tukey's test, $P < 0.05$).

but not to the N-terminal LOV1 and LOV2 domains (FLAG-P1N) (Fig. 4m), indicating that the kinase domain mediates the interaction with WDR48. From these results, we concluded that WDR48 is a direct target of phototropin kinases.

## WDR48 interacts with phot1 at the plasma membrane and chloroplast envelope

The subcellular localisation of phot1-GFP and GFP-WDR48 in guard cells was observed using confocal laser scanning microscopy. In the dark, GFP fluorescence of phot1-GFP was mainly observed at the plasma membrane, with weaker signals occasionally detectable around chloroplasts; meanwhile, blue light irradiation enhanced phot1-GFP signals in the cytoplasm (Fig. 5a, b). In contrast, the GFP-WDR48 signals were partially aggregated and distributed in the cytoplasm, with additional signals detected along the plasma membrane and the chloroplast envelope, and blue light irradiation did not significantly alter its localisation patterns (Fig. 5a, b and Supplementary Fig. 8). Furthermore, neither the phosphodefective nor the phosphomimetic mutations altered the localisation of GFP-WDR48 (Supplementary Fig. 9). To elucidate the localisation of phot1-GFP and GFP-WDR48 on the chloroplast envelope more precisely, we introduced the chloroplast envelope marker OUTER ENVELOPE MEMBRANE PROTEIN 7 (OEP7)-mCherry into plants expressing either phot1-GFP or GFP-

WDR48[40]. Analysis revealed that signals from both proteins were detectable on the chloroplast envelope under both dark and blue light conditions, as confirmed by colocalisation with OEP7-mCherry (Supplementary Fig. 10).

We further visualised the interaction between phot1 and WDR48 in plants using a bimolecular fluorescence complementation (BiFC) assay. When WDR48 fused to the N-terminal half of YFP (YFPn-WDR48) was co-expressed with phot1 fused to the C-terminal half of YFP (YFPc-phot1) in mesophyll cell protoplasts, YFP signals were detected at the cell periphery and co-localised with the signals of POTASSIUM CHANNEL IN ARABIDOPSIS THALIANA 1 (KAT1)-mCherry, a plasma membrane marker (Fig. 5c)[41]. In addition, when the depth of focus was changed, YFP signals were detected around the chloroplasts, where they co-localised with OEP7-mCherry (Fig. 5e). Furthermore, to confirm the interaction between phot1 and WDR48 in the BiFC assay, we performed a BiFC competition assay in which non-fused interacting proteins inhibited the BiFC reaction[42]. A decrease in the YFP signal was detected when 3×FLAG-WDR48 was co-expressed as a competitor for the BiFC reaction (Fig. 5c–f). We also confirmed that no YFP signals were observed when PIN-FORMED1 (PIN1) fused to the C-terminal half of YFP (YFPc-PIN1) was co-expressed with YFPn-WDR48 as a negative control (Fig. 5c). These findings suggest that WDR48 and phot1 interact at the plasma membrane and chloroplast envelope.

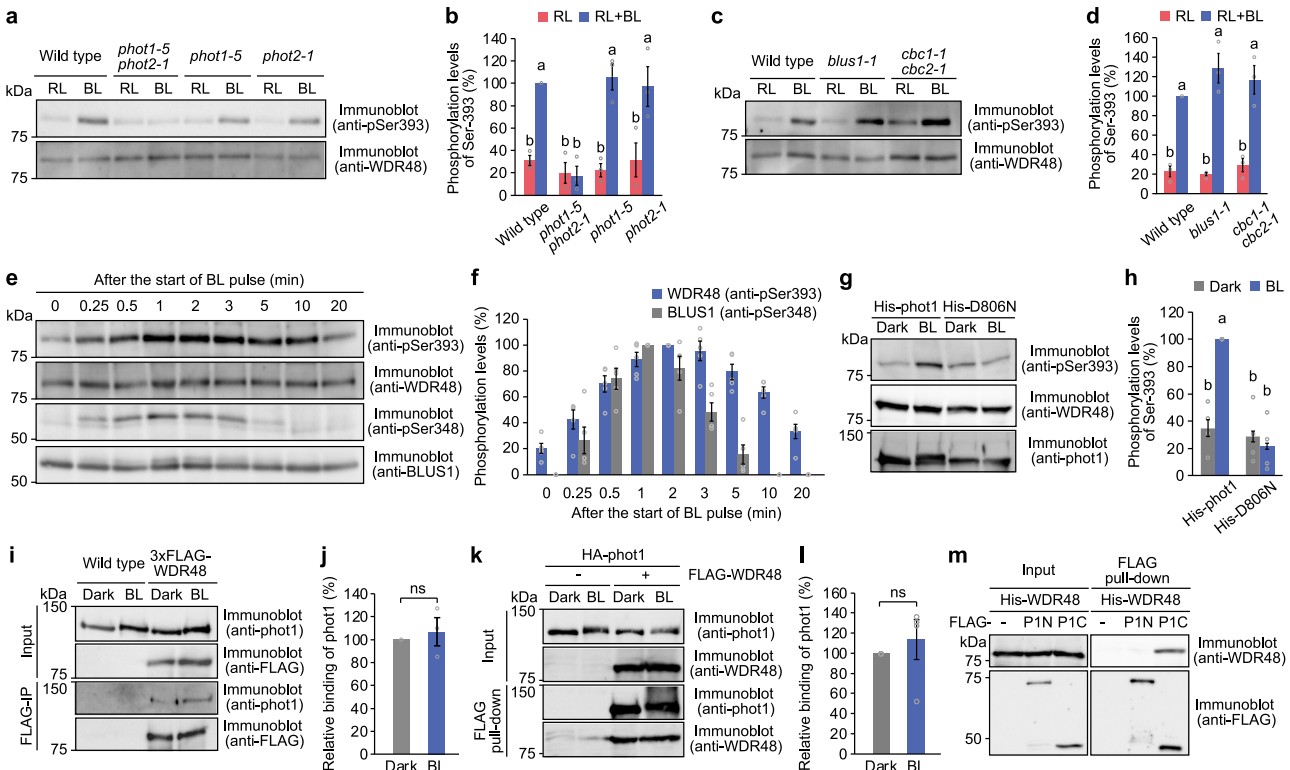

**Fig. 4 | Phosphorylation of WDR48 by phototropin kinases. a**, **b** Blue light-dependent phosphorylation of WDR48 in phototropin mutants. Guard cell protoplasts were pre-illuminated with red light (RL: 300 µmol m⁻² s⁻¹) for 30 min, after which a pulse of blue light (BL: 100 µmol m⁻² s⁻¹, 30 s) was superimposed on the background of RL. Phosphorylation of WDR48 was detected by immunoblotting with anti-pSer393 of WDR48 antibodies. **c**, **d** BL-dependent phosphorylation of WDR48 in the *blus1* and *cbc1 cbc2* mutants. **e**, **f** Time course of the BL-dependent phosphorylation of WDR48 and BLUS1. Guard cell protoplasts were pre-illuminated with RL (300 µmol m⁻² s⁻¹) for 30 min, after which a pulse of BL (100 µmol m⁻² s⁻¹, 30 s) was superimposed on the background of RL. The reaction was terminated at the indicated times after the start of BL illumination. Phosphorylation of WDR48 and BLUS1 was detected using anti-pSer393 of WDR48 and anti-pSer348 of BLUS1, respectively. **g**, **h** In vitro kinase assay. Recombinant His-phot1 and the corresponding kinase-dead mutants of phot1 (D806N) were incubated with FLAG-WDR48 in the presence of ATP for 1 h under dark or BL conditions. **i**, **j** Co-

immunoprecipitation assay. Microsomal fractions were prepared from etiolated seedlings of wild-type and transgenic plants expressing 3×FLAG-WDR48, which had been subjected to dark treatment or BL illumination, and then incubated with anti-FLAG agarose beads. **k**, **l** In vitro pull-down assay of FLAG-WDR48 and HA-phot1. Pull-downs were performed using anti-FLAG agarose beads in the presence or absence of BL. **m** In vitro pull-down assay of His-WDR48 with FLAG-tagged N- and C-terminal fragments of phot1, performed using anti-FLAG agarose beads. For (**b**) and (**d**), data represent mean ± SEM (*n* = 3 biologically independent samples; One-way ANOVA with Tukey's test, *P* < 0.01). For (**f**), data represent mean ± SEM (*n* = 5 biologically independent samples). For (**h**), data represent mean ± SEM (*n* = 4 independent experiments; One-way ANOVA with Tukey's test, *P* < 0.01). For (**j**), data represent mean ± SEM (*n* = 3 biologically independent samples). For (**l**), data represent mean ± SEM (*n* = 4 independent experiments). ns indicates no significant difference (Student's *t*-test, *P* < 0.01).

## BLUS1 and WDR48 coordinately regulate starch degradation in response to blue light

Blue light signalling mediated by BLUS1 activates the plasma membrane H⁺-ATPase, which is required for starch degradation during stomatal opening (Supplementary Fig. 6c, d)[30]. Considering that the blue light activation of H⁺-ATPase was not affected in the *wdr48* mutant, WDR48 is most likely not involved in signalling upstream of H⁺-ATPase activation. Furthermore, phosphorylation of WDR48 was induced by blue light but not by the application of fusicoccin (Fc), an activator of H⁺-ATPase (Supplementary Fig. 11a, b). In addition, blue light-dependent WDR48 phosphorylation was observed in the *blus1-1* and *aha1-9* mutants, as in the wild type (Fig. 4c, d; Supplementary Fig. 11c, d), suggesting that WDR48 does not function downstream of H⁺-ATPase activation. Thus, WDR48 appears to constitute a separate signalling pathway from that of H⁺-ATPase activation and induce starch degradation.

To further confirm that WDR48 induces starch degradation via a pathway distinct from the activation of H⁺-ATPase, we examined starch degradation by Fc. A previous study indicated that treatment with 10 µM Fc induced starch degradation in guard cells in the isolated epidermis, regardless of blue light irradiation[30]. Consistent with this finding, treatment with 10 µM Fc under red light induced starch degradation and

stomatal opening in the wild-type as well as in the *wdr48* mutants (Supplementary Fig. 12a–c). However, treatment with such a high concentration (10 µM) of Fc caused a two-times higher rate of H⁺ pumping compared to blue light conditions (Supplementary Fig. 12d) and induced stomatal opening even in the dark without starch degradation (Supplementary Fig. 12e–g). However, when epidermal peels were treated with 1 µM Fc, which induced a similar rate of H⁺ pumping to that of blue light (Supplementary Fig. 12d), neither starch degradation nor stomatal opening was observed under red light or in the dark (Supplementary Fig. 12a–c, e–g). These results suggest that activation of H⁺-ATPase alone is not sufficient to trigger starch degradation and stomatal opening under physiological conditions and that additional blue light signals are required for the induction of these responses.

Next, we investigated starch degradation and stomatal opening by blue light and 1 µM Fc in *blus1* and *wdr48* single mutants and *blus1 wdr48* double mutant. If BLUS1 and WDR48 function in separate signalling pathways and both pathways are required for starch degradation, we would expect that the application of 1 µM Fc, together with blue light, would induce starch degradation and stomatal opening in the *blus1* mutant, but not in the *wdr48* or *blus1 wdr48* mutants. In the wild-type, blue light irradiation or simultaneous application of blue

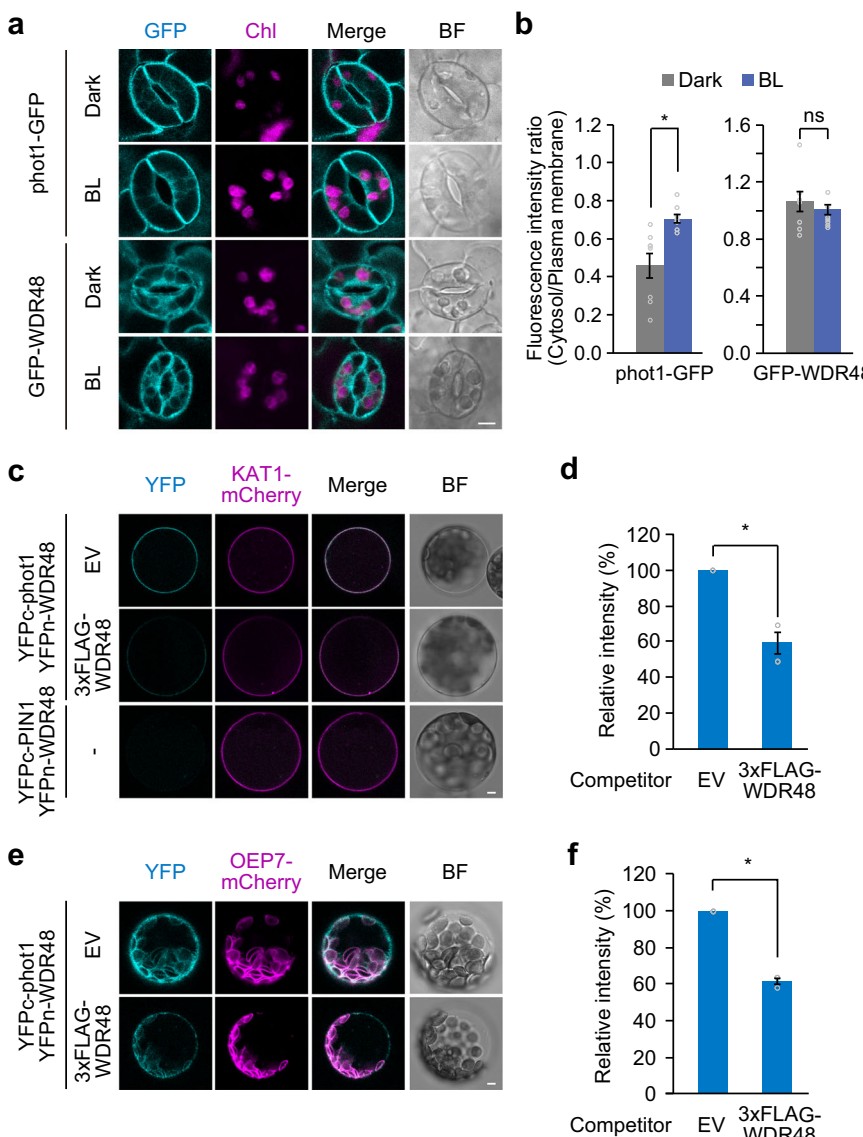

**Fig. 5 | Subcellular localisation and in vivo interaction between phot1 and WDR48.** Confocal images of phot1-GFP and GFP-WDR48 in guard cells under dark and blue light (BL) conditions (**a**) and the cytosol/plasma membrane fluorescence intensity ratio (**b**). Leaves from dark-adapted plants were illuminated with blue light (BL: 10 μmol m⁻² s⁻¹) for 10 min. GFP, GFP fluorescence; Chl, chlorophyll fluorescence; Merge, merged image of GFP and Chl fluorescence; BF, bright field. Bar represents 5 μm. BiFC assay (**c**, **e**) and the relative YFP fluorescence intensity (**d**, **f**). The N- or C-terminal fragments of YFP were fused with WDR48 (YFPn-WDR48) or phot1 (YFPc-phot1) and co-expressed in mesophyll cell protoplasts by PEG-calcium transfection, together with KAT1-mCherry or OEP7-mCherry as plasma membrane and chloroplast envelope markers, respectively. After transfection, the protoplasts were kept in the dark prior to imaging, which was performed using a confocal laser-scanning microscope. **c**, **e** Represent the fluorescence images of different focal planes. YFPn-PIN1 was used as the negative control, and 3×FLAG-WDR48 was co-expressed as a competitor. YFP, YFP fluorescence; Merge, merged image of YFP and mCherry; EV, empty vector. Bars represent 5 μm. For (**b**), data represent mean ± SEM (n = 8 biologically independent cells). For (**d**) and (**f**), data represent mean ± SEM (n = 3 independent experiments). Asterisk indicates significant difference (Student's *t*-test, *P* < 0.01). ns indicates no significant difference.

light and Fc induced starch degradation and stomatal opening, but Fc alone did not induce either response (Fig. 6a–c). In contrast, blue light and Fc did not stimulate starch degradation or stomatal opening in the *wdr48* mutant under any condition. Similarly, in the *blus1* mutant, which lacks blue light-dependent activation of H⁺-ATPase[22], neither blue light nor Fc alone induced starch degradation or stomatal opening, whereas the combination of blue light and Fc elicited both responses. However, blue light- and Fc-induced starch degradation and stomatal opening observed in the *blus1* mutant were not found in the *blus1 wdr48* double mutant.

Assuming that both H⁺-ATPase activation and WDR48 phosphorylation are required for starch degradation, the phosphodefective WDR48 variant would not exhibit starch degradation even when H⁺-ATPase is activated by Fc, because WDR48 cannot be phosphorylated. In contrast, the phosphomimetic WDR48 variant is expected to trigger starch degradation if H⁺-ATPase is activated by Fc. To test this, we examined starch degradation and stomatal opening in transgenic lines expressing either phosphodefective (GFP-S393A) or phosphomimetic (GFP-S393E) variants of WDR48, alongside the GFP-WDR48 line and the *wdr48* mutant. As expected, starch degradation and stomatal opening were induced by blue light or by the combination of blue light and Fc in the GFP-WDR48 line (Fig. 6d–f). In contrast, neither response was observed under any treatment in the phosphodefective GFP-S393A line or in the *wdr48* mutant. Notably, in the phosphomimetic GFP-S393E line, Fc treatment alone was sufficient to induce both starch degradation and stomatal opening in the absence of blue light.

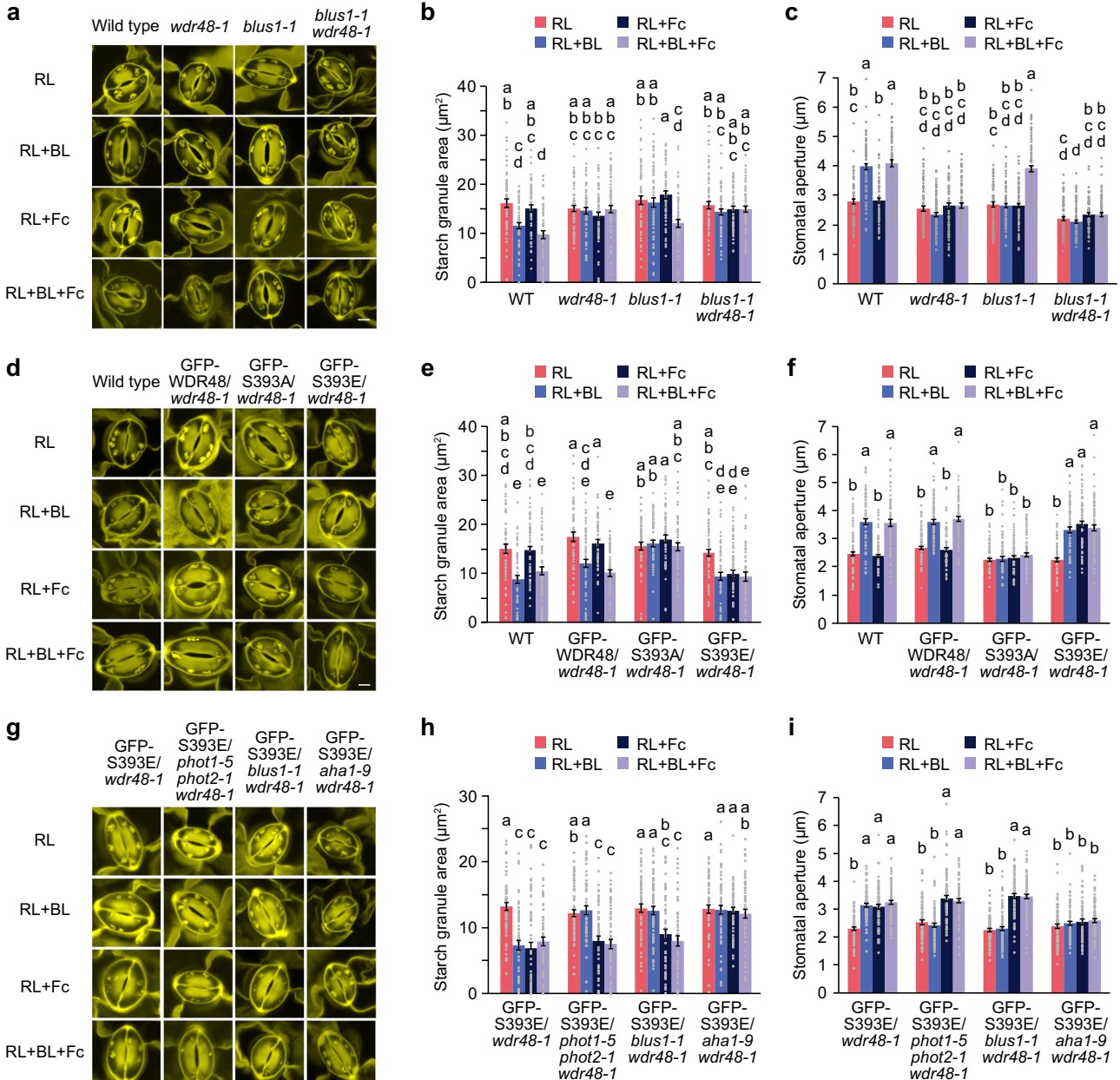

**Fig. 6 | Coordinated control of blue light-induced starch degradation and stomatal opening by BLUS1 and WDR48.** Light- and fusicoccin (Fc)-induced changes in starch dynamics (**a**, **b**) and stomatal opening (**c**) in the *blus1* and *wdr48* mutants. Epidermal stripes from dark-adapted leaves were pre-illuminated with red light (RL: 50 μmol m$^{-2}$ s$^{-1}$) for 2 h, after which blue light (BL: 10 μmol m$^{-2}$ s$^{-1}$) was superimposed on the background of RL for 10 min or 1 μM Fc was added to the epidermis with or without BL illumination. Light- and Fc-induced changes in starch dynamics (**d**, **e**) and stomatal opening (**f**) in transgenic plants expressing

phosphodefective and phosphomimetic WDR48. Light- and Fc-induced changes in starch dynamics (**g**, **h**) and stomatal opening (**i**) in transgenic plants expressing phosphomimetic WDR48 in *phot1 phot2*, *blus1*, and *aha1* mutant backgrounds. For (**a**, **d**, **g**), bars represent 5 μm. For (**b**), (**e**), and (**h**), data represent mean ± SEM ($n = 60$ stomata examined over three independent experiments; One-way ANOVA with Tukey's test, $P < 0.05$). For (**c**, **f**, **i**), data represent mean ± SEM ($n = 75$ stomata examined over three independent experiments; One-way ANOVA with Turkey's test, $P < 0.01$).

In transgenic lines expressing the phosphomimetic GFP-S393E in *phot1 phot2 wdr48* or *blus1 wdr48* mutant backgrounds, blue light alone did not induce starch degradation or stomatal opening, which is consistent with impaired H$^+$-ATPase activation in these mutants (Fig. 6g–i). However, when Fc was applied, both responses were successfully triggered. In the GFP-S393E expressing line in the *aha1 wdr48* background, blue light alone did not induce starch degradation or stomatal opening, and neither Fc nor blue light in combination with Fc induced these responses. These findings suggest that phosphorylation of WDR48, together with BLUS1-mediated H$^+$-ATPase activation is essential for starch degradation and stomatal opening.

## Discussion

Using a combination of biochemical, reverse genetics, and quantitative imaging approaches, we identified the WD-repeat protein WDR48 as a critical regulator of blue light-dependent starch degradation and stomatal opening. Phosphoproteomic analysis revealed that WDR48 undergoes rapid phosphorylation in guard cells in response to blue light (Fig. 1a, b). Loss of *WDR48* abolished blue light-dependent starch degradation without affecting red light-induced starch accumulation, and impaired stomatal opening to blue light (Fig. 2c–g). These findings suggest that WDR48 mediates blue light-dependent starch degradation in guard cells, thereby contributing to stomatal opening.

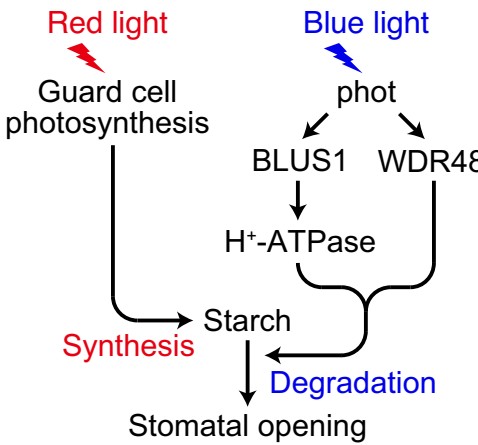

**Fig. 7 | Model of synergistic stomatal opening via red light-dependent starch synthesis and blue light-dependent starch degradation.** Red light promotes starch accumulation in guard cell chloroplasts through photosynthesis. Blue light activates phototropins, which phosphorylate both BLUS1 and WDR48. Phosphorylation of BLUS1 activates the plasma membrane H⁺-ATPase, and along with WDR48 phosphorylation, these signals lead to starch degradation, thereby promoting stomatal opening.

Site-directed mutagenesis demonstrated that phosphorylation of WDR48 at Ser-393 is essential for its function in blue light-dependent starch degradation and stomatal opening (Fig. 3). Immunoblot analysis with phospho-specific antibodies showed that this phosphorylation depends on phot1 and phot2, but occurs independently of BLUS1 and CBC1/CBC2 (Fig. 4a–d). Consistent with this, phot1 and phot2 directly phosphorylated WDR48 at Ser-393 in vitro (Fig. 4g, h and Supplementary Fig. 7a, b), and WDR48 physically interacted with phot1 in vitro and in vivo (Fig. 4i–m). These findings establish WDR48 as a direct substrate of phototropin kinases.

BLUS1-mediated H⁺-ATPase activation has been shown to drive starch degradation in response to blue light, thereby promoting stomatal opening[30]. Consistent with this, blue light signalling mutants, including *phot1 phot2*, *blus1*, and *aha1*, exhibited impaired starch degradation by blue light (Fig. 2c–f and Supplementary Fig. 6). However, WDR48 did not affect blue light-dependent H⁺-ATPase activation (Fig. 1i, j), and its phosphorylation was not induced by the H⁺-ATPase activator Fc (Supplementary Fig. 11a, b). Furthermore, experiments using moderate H⁺-ATPase activation with 1 μM Fc showed that although the *blus1* mutant lacks blue light-dependent H⁺-ATPase activation[22], Fc treatment restored starch degradation under blue light, and this effect was abolished by the *wdr48* mutation (Fig. 6a, b). Together, these results suggest that WDR48 defines an additional blue light signalling pathway for starch degradation that functions in parallel with BLUS1-mediated H⁺-ATPase activation.

The functional relationship between the two pathways was further clarified using phosphomimetic and phosphodefective WDR48 variants expressed in the *wdr48* mutant background. GFP-S393E induced starch degradation and stomatal opening upon Fc-induced H⁺-ATPase activation even in the absence of blue light (Fig. 6d–f), suggesting that WDR48 phosphorylation is sufficient to bypass the blue light requirement once the H⁺-ATPase is activated. However, these Fc-induced responses were abolished in the *aha1 wdr48* mutant background (Fig. 6g–i). Moreover, although GFP-S393E showed blue light-dependent starch degradation and stomatal opening, these responses were lost in the *phot1 phot2 wdr48* or *blus1 wdr48* mutant backgrounds (Fig. 6g–i), confirming that WDR48 phosphorylation alone is insufficient without BLUS1-mediated H⁺-ATPase activation. In contrast, the phosphodefective S393A variant was completely inactive under all conditions (Fig. 6d–f), underscoring the critical role of WDR48 phosphorylation. Together, these findings support a model in which blue light-dependent starch degradation integrates two signalling pathways: WDR48-mediated signalling and BLUS1-mediated activation of the plasma membrane H⁺-ATPase (Fig. 7).

The mechanisms by which WDR48 promotes starch degradation remain to be elucidated. In animals, WDR48 binds to deubiquitinating enzymes (DUBs) and enhances their enzymatic activity[43]. Arabidopsis contains 27 DUB genes that participate in diverse physiological processes[44–46]. Pharmacological inhibition of DUBs suppressed light-induced stomatal opening[47], and mutation of Arabidopsis *UBIQUITIN-SPECIFIC PROTEASE 24* (*UBP24*), a member of the DUB family, impairs abscisic acid-induced stomatal closure[48]. Given these findings, WDR48 may influence starch degradation by modulating DUB activity in guard cells, and this possibility warrants further investigation.

H⁺-ATPase activity may promote starch degradation through cytosolic alkalinisation, which enhances amylase activity[30]. In addition, TARGET of RAPAMYCIN (TOR) kinase regulates guard cell starch degradation and stomatal opening by controlling BAM1 stability and expression[49]. Notably, in yeast, TOR is activated through H⁺-coupled nutrient uptake driven by the plasma membrane H⁺ gradient generated by the H⁺-ATPase[50]. Further studies are required to clarify how WDR48- and BLUS1-mediated signalling converge to coordinate starch degradation in guard cells.

Our BiFC assay revealed that WDR48 interacts with phot1 at both the plasma membrane and chloroplast envelope (Fig. 5c, e). The functional significance of these localisations in starch degradation remains to be determined. Targeted localisation studies could shed light on this mechanism. Furthermore, the *wdr48* mutant displayed impaired red light-dependent stomatal opening, which was complemented by both phosphodefective and phosphomimetic WDR48 forms (Fig. 3d, f), suggesting potential phototropin-independent roles of WDR48 in stomatal opening.

Phototropins phosphorylate multiple substrates, including BLUS1 and CBC1, to coordinate membrane hyperpolarisation and ion transport during stomatal opening[22,27]. Our study identified WDR48 as an additional phototropin substrate that stimulates starch degradation within guard cells, in conjunction with BLUS1. Collectively, these findings suggest that phototropins establish a signalling network by phosphorylating multiple targets to integrate guard cell responses. Identification of further phototropin substrates will advance our understanding of these regulatory networks.

In conclusion, this study establishes WDR48 as a key phototropin substrate that contributes to blue light-dependent starch degradation in guard cells. Red light-driven guard cell photosynthesis promotes starch accumulation that subsequently serves as the substrate for blue light-dependent degradation. In parallel with BLUS1-mediated H⁺-ATPase activation, WDR48 regulates these starch dynamics, providing a mechanistic basis for the synergistic effect of red and blue light on stomatal opening (Fig. 7). These findings provide a framework for understanding how phototropin-mediated signalling integrates with photosynthesis-derived cues to coordinate guard cell function.

## Methods

### Plant materials and growth conditions

*Arabidopsis thaliana* accession Columbia-0 (Col-0) was used as the wild-type. *phot1-5 phot2-1*[10], *phot1-5*[51], *phot2-1*[52], *blus1-1*[22], *aha1-9*[35], and *cbc1-1 cbc2-1*[27] have been described previously. T-DNA insertion mutants of *wdr48-1* (GABI_585_E01) and *wdr48-2* (SALK_059570C) were obtained from the Nottingham Arabidopsis Stock Centre (NASC). The transgenic plants of phot1-GFP have been described previously[22]. Plants were grown on a soil:vermiculite mixture (1:1) for 4 weeks under white light (50 μmol m⁻² s⁻¹) with 14 h/10 h light/dark cycles at 24 °C.

### Phosphoproteomic analysis

Phosphoproteomic analysis was conducted using guard cell protoplasts from the wild-type and *phot1-5 phot2-1* mutants[17,22,27]. Guard cell

protoplasts were incubated under red light (600 µmol m$^{-2}$ s$^{-1}$) for 30 min, after which a pulse of blue light (100 µmol m$^{-2}$ s$^{-1}$, 30 s) was superimposed on the background of red light. The reaction was terminated 2.5 min after the start of blue light illumination by the addition of trichloroacetic acid to the protoplast suspension. Following digestion with Lys-C and trypsin, the phosphopeptides were enriched by hydroxy acid-modified metal oxide chromatography (HAMMOC) using lactic acid-modified titania[53,54]. The enriched phosphopeptides were analysed by nanoLC/MS/MS in Data Dependent Acquisition (DDA) mode using LTQ-Orbitrap (Thermo Fisher Scientific)[17,22,27] or Orbitrap Fusion Lumos (Thermo Fisher Scientific) under the following conditions.

The Orbitrap Fusion Lumos was coupled with Ultimate3000 pump (Thermo Fisher Scientific) and an HTC-PAL autosampler (CTC Analytics). Reprosil-Pur 120 C18-AQ beads (1.9 µm; Dr. Maisch GmbH) packed in a self-pulled needle (150 mm length × 100 µm i.d., 6-µm opening) were used as the nanoLC column. The mobile phases consisted of 0.5% acetic acid (A) and 0.5% acetic acid and 80% acetonitrile (B). A three-step linear gradient of 5% to 10% B in 5 min, 10% to 40% B in 60 min, 40% to 100% B in 5 min, and 100% B for 10 min was applied at a flow rate of 500 nL/min. A spray voltage of 2400 V was applied, and the FAIMS compensation voltage was set to −40, −60, and −80. The MS scan range was $m/z$ 300–1500. MS scans were performed using the Orbitrap with $r = 120,000$, and subsequent MS/MS scans were performed by the quadrupole ion trap. Synchronous precursor selection (SPS) was employed with a higher-energy collisional dissociation (HCD) at 10 SPS precursors. The HCD collision energy was set to 30.

The raw LTQ-Orbitrap data were analysed with Mascot version 2.3 (Matrix Science) against the TAIR database (release 10) as previously described[17,22,27]. The raw Orbitrap Fusion Lumos data were analysed with MSFragger (version 4.1)[55] within the FragPipe platform (v22.0) against the TAIR database (release 10) as follows. Mass tolerances of 10 ppm and 0.4 Da were applied to precursor and fragment ions, respectively, and mass calibration and parameter optimisation were enabled. Trypsin was specified as the reference enzyme, and up to two missed cleavages were allowed. Carbamidomethylation of cysteine was set as a fixed modification, and oxidation of methionine and phosphorylation of serine, threonine and tyrosine were allowed as variable modifications. PeptideProphet and ProteinProphet in Philosopher[56] were used to filter the results with FDR < 1%. The identified phosphopeptides were requantified using Slyline[57].

## Isolation of guard cell protoplasts and measurement of H$^+$ pumping

Guard cell protoplasts were enzymatically isolated from the leaves[25,58]. The leaves of 4-week-old plants were homogenised using a blender (Warning), and epidermal fragments were collected on a 58 µm nylon mesh. The epidermal fragments were incubated in an enzyme solution containing 10 mM MES-KOH (pH 5.4), 0.25 M mannitol, 0.5% (w/v) cellulase R-10, 0.05% (w/v) macerozyme R-10, 0.1% (w/v) polyvinylpyrrolidone K-30, 0.2% (w/v) bovine serum albumin, and 1 mM CaCl$_2$ for 30 min at 24 °C. The fragments were then recovered on the nylon mesh and washed with 0.3 M mannitol and 1 mM CaCl$_2$. Subsequently, the epidermal fragments were incubated in a second enzyme solution containing 10 mM MES-KOH (pH 5.4), 0.4 M mannitol, 1.5% (w/v) cellulase RS, 0.5% (w/v) macerozyme R-10, 0.2% (w/v) bovine serum albumin, and 1 mM CaCl$_2$ for 1 h at 27 °C. The guard cell protoplasts were filtered sequentially through 94 µm, 25 µm, and four layers of 10 µm nylon meshes and collected by centrifugation at 420 × $g$ for 18 min. The protoplasts were washed twice with 0.4 M mannitol and 1 mM CaCl$_2$ and used for analysis. To measure H$^+$ pumping, guard cell protoplasts were incubated in 0.125 mM MES-NaOH (pH 6.0), 1 mM CaCl$_2$, 0.4 M mannitol, and 10 mM KCl under red light (300 µmol m$^{-2}$ s$^{-1}$) for 2 h, after which a pulse of blue light

(100 µmol m$^{-2}$ s$^{-1}$) was superimposed on the background of red light for 30 s. H$^+$ pumping was measured using a pH electrode.

## Immunoblot analysis

Guard cell protoplasts were incubated in 0.125 mM MES-NaOH (pH 6.0), 1 mM CaCl$_2$, 0.4 M mannitol, and 10 mM KCl under red light (300 µmol m$^{-2}$ s$^{-1}$) for 30 min, and then a pulse of blue light (100 µmol m$^{-2}$ s$^{-1}$) was superimposed on the background of red light for 30 s. The reaction was terminated at 2 min for phot1, phot2, WDR48, and BLUS1 and 3 min for H$^+$-ATPase after the start of irradiation with a blue light pulse by adding trichloroacetic acid to guard cell protoplasts. After washing twice with 50 mM Tris-HCl (pH 8.0), the precipitated proteins were resuspended in the SDS sample buffer and subjected to SDS-PAGE.

For λ-PPase treatment, guard cell protoplasts were incubated as described above and collected by centrifugation at 10,000 × $g$ for 15 s. Pellets were resuspended in buffer containing 50 mM HEPES (pH 7.5), 100 mM NaCl, 2 mM dithiothreitol (DTT), 0.01% (w/v) Brij 35, 1 mM MnCl$_2$, 0.4% (v/v) Triton X-100, 0.5 mM phenylmethylsulfonyl fluoride (PMSF), 10 µM leupeptin, and 400 units of λ-PPase (New England Biolabs, P0753), and incubated at 30 °C for 60 min. After the reaction, SDS sample buffer was added to the λ-PPase-treated guard cell proteins, and the samples were subjected to SDS-PAGE.

Immunoblot analysis was performed as previously reported[25,59]. Antibodies against WDR48 and phosphorylated Ser-393 of WDR48 were produced by immunising rabbits with recombinant GST-WDR48 (V300-R753) and synthetic phospho-peptide NRARVpSLEGLNPA, respectively. Antibodies against phot1, phot2, BLUS1, pSer348 of BLUS1, H$^+$-ATPase, and pThr948 of AHA1 have been previously described[17,22,60,61]. Anti-GFP and anti-FLAG-HRP antibodies were purchased from Invitrogen (A-6455) and Sigma-Aldrich (A8592), respectively. Antibodies were used at the following dilutions: 1:1000 for anti-WDR48, anti-pSer393 of WDR48, and anti-pSer348 of BLUS1; 1:3000 for anti-BLUS1, anti-GFP, anti-phot1, and anti-phot2; 1:5000 for anti-pThr948 of AHA1 and anti-FLAG-HRP; and 1:8000 for anti-H$^+$-ATPase.

## Measurement of stomatal opening

The stomatal opening in the isolated epidermis was measured as described previously[10], with slight modifications. Epidermal strips from leaves of dark-adapted plants were incubated in stomatal opening buffer containing 5 mM MES-bistrispropane (pH 6.5), 50 mM KCl, and 0.1 mM CaCl$_2$ under red light (50 µmol m$^{-2}$ s$^{-1}$) and blue light (10 µmol m$^{-2}$ s$^{-1}$) for 2 h at 24 °C. For the measurements of rapid stomatal opening, the epidermal strips were preirradiated with red light (50 µmol m$^{-2}$ s$^{-1}$) for 2 h, after which blue light (10 µmol m$^{-2}$ s$^{-1}$) was superimposed on the background of red light or Fc (1 or 10 µM) was added to the buffer, and further incubated for the indicated time. Images of stomata in the abaxial epidermis were acquired using an inverted microscope (Eclipse TS100; Nikon). The stomatal aperture was determined using the ImageJ 1.53q software (National Institutes of Health).

Stomatal conductance in intact leaves was measured using a gas-exchange measurement system (LI-6400; Li-COR), as described previously, with settings of 350 ppm CO$_2$, 40–60% relative humidity, 24 °C leaf temperature, and 200 µmol m$^{-1}$ flow rate[62]. Leaves of dark-adapted plants were irradiated with red light (300 µmol m$^{-2}$ s$^{-1}$) for 1 h, after which blue light (10 µmol m$^{-2}$ s$^{-1}$) was superimposed on the background red light for 20 min.

## PS-PI stain and quantification of guard cell starch

Abaxial epidermal peels from dark-adapted plants were incubated as described for the measurement of stomatal opening. Guard cell starch granules were fixed and stained with PS-PI staining[30,38]. Fluorescence images were obtained using a confocal laser scanning microscope

(488 nm excitation and 590 nm long pass emission) (Digital Eclipse C1; Nikon).

To quantify the guard cell starch area, we used the X-net[63] for automatic starch segmentation. X-net is a cell image segmentation method consisting of two encoders and two decoders, and performed better than U-net[64]. In X-net, the input image is fed into two encoders, and two feature maps in two encoders are aggregated by concatenation. This allows us to extract rich features that cannot be extracted by a single network and improves the segmentation accuracy. To train and evaluate the X-net for starch segmentation, we used 319 images consisting of 2 classes: starch and others. The 319 images were annotated manually using the ImageJ 1.53q software. We cropped the regions of $512 \times 512$ pixels from the original images to train the X-net because of the memory of the GPU. We used Intersection over Union (IoU), which is the overlap ratio between the segmentation result and manual annotation, as an evaluation measure. X-net was evaluated using four-fold cross-validation. The IoUs of the starch and other classes were 69.75% and 99.63%, respectively. The starch granule area was determined by dividing the labelled pixel area by the pixel density (pixels $\mu m^{-2}$) of the analysed image.

## Construction of transgenic plants

The 7370 bp genomic sequence of *WDR48* containing the promoter, coding, and 3′-UTR region was amplified using primers 5′-GCTTGA-TATCGAATTCCGCTTAAGGAAACTAGTGAGTG-3′ and 5′-TAGAACTAG TGGATCCGAGAAGCTAAGTAAACACTAGGC-3′. The resulting product was subcloned into *Bam*H I/*Eco*R I site of the pBluescript SK (+) vector (Stratagene) using the In-Fusion HD Cloning Kit (Clontech). Site-directed mutagenesis was performed using a QuikChange Site-Directed Mutagenesis Kit (Stratagene). The primers used were 5′-CTTTTAA-TAGGGCGAGGGTAGCTTTGGAAGGACTAAATCC-3′ and 5′-GGATTTAG TCCTTCCAAAGCTACCCTCGCCCTATTAAAAG-3′ for *WDR48-S393A* and 5′-CTTTTAATAGGGCGAGGGTAGAATTGGAAGGACTAAATCC-3′ and 5′-GGATTTAGTCCTTCCAATTCTACCCTCGCCCTATTAAAAG-3′ for *WDR48-S393E*. To construct the *pWDR48:GFP-WDR48* vector, the kanamycin resistance gene, *NPT II*, in pRI 101-AN (TaKaRa), was replaced with the hygromycin resistance gene, *HPT*, and the resulting vector was designated pPCS201. The *GFP* sequence was then inserted into the *Sal* I/*Bam*H I sites to generate pPCS202. The upstream 1398 bp sequence, including the promoter region of *WDR48*, was amplified using primers 5′-GGCCAGTGCCAAGCTTGCTTAAGGAAACTAGTGAGTG-3′ and 5′-TACCCCCGGGGTCGACTTCAAATTATTCTGAAAGCCTGAAAGGAG-3′, and the product was subcloned into the *Hind* III/*Sal* I sites of pPCS202. After amplifying the downstream 5,981 bp sequence, including the coding and 3′-UTR regions of *WDR48* using primers 5′-CAAGGGATCC-GAATTCATGCACCGGGTTGGAAGTGC-3′ and 5′-TGTTGATTCA-GAATTCGAGAAGCTAAGTAAACACTAGGC-3′, the resulting product was inserted into the *Eco*R I site of pPCS202 containing the upstream sequence. To construct the *p35S:3×FLAG-WDR48* vector, the *3×FLAG* sequence was inserted into the *Sal* I/*Bam*H I sites of pPCS201, and the resulting vector was designated pPCS206. The 5981 bp genomic sequence, including the coding and 3′-UTR regions of *WDR48*, was amplified using primers 5′-CGATGACAAGGGATCCATGCACCGG GTTGGAAGTGC-3′ and 5′-TTCAGAATTCGGATCCGAGAAGCTAA GTAAACACTAGGC-3′, and the resulting product was subcloned into *Bam*H I site of pPCS206. Each construct was transformed into the *wdr48-1* mutant via floral dip using *Agrobacterium tumefaciens* strain GV3101.

To construct the *pGC1:OEP7-mCherry* vector, the *NOS* promoter and *NPT II* in pRI 101-AN were replaced with *pOLE1:OLE1-TagRFP*[65], generating pPCS501. The *35S* promoter of pPCS501 was then replaced with the *GC1* promoter to yield pPCS601. *OEP7* fused to mCherry was amplified using primers 5′-AAGAAATATGGTCGACATGGGAAA AACTTCGGGAGCG-3′ and 5′-TTCAGAATTCGGATCCTTACTTGTA-CAGCTCGTCCATG-3′, and the product was subcloned into the *Sal* I/

*Bam*H I sites of pPCS601. The construct was introduced into phot1-GFP- or GFP-WDR48-expressing plants.

## In vitro pull-down assay

FLAG-WDR48, His-WDR48, FLAG-P1N (N-terminal fragment of phot1, M1-N619), and FLAG-P1C (C-terminal fragment of phot1, M620-F996) were synthesised using an in vitro transcription/translation kit[66] (Bio-Sieg). To express HA-phot1, cDNA sequence of *PHOT1* fused to an HA tag was amplified using primers 5′-TATAGAATACAAGCTTATGT ATCCTTACGATGTTCCAGATTATGCTATGGAACCAACAGAAAAACCAT CGAC-3′ and 5′-GCTCGCCCGGGGATCCTCAAAAAAACATTTGTTTGCA-GATCTTCTAGCTC-3′, and the product was subcloned into the *Hind* III/ *Bam*H I sites of pSP64 poly(A) vector (Promega, P1241). The resulting construct was added to the reaction mixture of the TnT SP6 High-Yield Wheat Germ Protein Expression System (Promega, L3260), supplemented with $10 \mu M$ flavin mononucleotide (FMN), as described previously[67].

For the in vitro pull-down assay using full-length phot1, FLAG-WDR48 was diluted in buffer containing 20 mM Tris-HCl (pH 7.4), 140 mM NaCl, and 0.1% (v/v) Trinton X-100, and incubated with anti-FLAG M2 affinity gel (Sigma-Aldrich, A2220) at 4 °C for 1 h. After washing the beads three times, HA-phot1, diluted in buffer containing 37.5 mM Tris-HCl (pH 7.5), 5.3 mM $MgSO_4$, 90 mM NaCl, 1 mM EGTA, 1 mM DTT, $100 \mu M$ ATP, and $10 \mu M$ FMN, was added and incubated at 24 °C for 3 h in the dark or under blue light ($100 \mu mol \, m^{-2} s^{-1}$). After washing the beads three times, the bound proteins were mixed with the SDS sample buffer and subjected to immunoblotting using anti-phot1 and anti-WDR48 antibodies.

For the in vitro pull-down assay using the N- and C-terminal fragments of phot1, FLAG-P1N or FLAG-P1C was diluted in buffer containing 20 mM Tris-HCl (pH 7.4), 140 mM NaCl, and 0.1% (v/v) Triton X-100, and incubated with anti-FLAG M2 affinity gel at 4 °C for 1 h. After washing the beads three times, His-WDR48, diluted in buffer containing 20 mM Tris-HCl (pH 7.4) and 140 mM NaCl, was added and incubated at 4 °C for 3 h in the dark. After washing the beads three times, the bound proteins were mixed with the SDS sample buffer and analysed by immunoblotting using anti-FLAG-HRP and anti-WDR48 antibodies.

## In vitro kinase assay

His-phot1, His-D806N, His-phot2, and His-D720N were synthesised using an in vitro transcription/translation kit[66] (BioSieg) supplemented with $10 \mu M$ FMN. The FLAG-WDR48 was synthesised using the same kit without FMN. His-tagged and FLAG-tagged proteins were incubated in buffer containing 37.5 mM Tris-HCl (pH 7.5), 5.3 mM $MgSO_4$, 90 mM NaCl, 1 mM EGTA, 1 mM DTT, $100 \mu M$ ATP, and $10 \mu M$ FMN at 24 °C for 1 h in the dark or under blue light ($100 \mu mol \, m^{-2} s^{-1}$). For λ-PPase treatment, FLAG-WDR48 was recovered from the reaction mixture using anti-FLAG M2 affinity gel and washed three times. The beads were then incubated with λ-PPase in buffer containing 50 mM HEPES (pH 7.5), 100 mM NaCl, 2 mM DTT, 0.01% (w/v) Brij 35, and 1 mM $MnCl_2$ at 30 °C for 1 h. The reaction mixtures were subjected to SDS-PAGE and analysed by immunoblotting with anti-pSer393 of WDR48, anti-WDR48, anti-phot1, and anti-phot2 antibodies.

## Co-immunoprecipitation

Three-day-old etiolated seedlings expressing 3×FLAG-WDR48, as well as wild-type seedlings, were kept in the dark or illuminated with blue light ($100 \mu mol \, m^{-2} s^{-1}$) for 2 min and then homogenised in extraction buffer containing 50 mM MOPS-KOH (pH 7.5), 2.5 mM EDTA, 100 mM NaCl, 0.5 mM PMSF, $10 \mu M$ leupeptin, 2 mM DTT, 10 mM NaF, 0.5 mM ammonium molybdate, and 100 nM calyclin A. After initial centrifugation at $10,000 \times g$ for 10 min at 4 °C, the supernatants were centrifuged at $100,000 \times g$ for 1 h at 4 °C to obtain microsomal membrane. The membrane was resuspended in an extraction buffer

containing 0.1% (v/v) Triton X-100 and kept on ice for 10 min. Following centrifugation at $10,000 \times g$ for 10 min, the supernatant was subjected to immunoprecipitation with anti-FLAG M2 affinity gel for 12 h at 4 °C. After washing the beads three times, the immunoprecipitated proteins were mixed with the SDS sample buffer and analysed by immunoblotting with anti-phot1 and anti-FLAG-HRP antibodies.

## Subcellular localisation

Leaves from 2-week-old dark-adapted plants expressing phot1-GFP, GFP-WDR48, GFP-S393A, or GFP-S393E, or co-expressing phot1-GFP and OEP7-mCherry or GFP-WDR48 and OEP7-mCherry, were illuminated with blue light (10 μmol m$^{-2}$ s$^{-1}$) for 10 min. GFP, mCherry, and chlorophyll fluorescence were then detected using a confocal laser scanning microscope (TCS SP8X; Leica Microsystems). For detection of GFP and mCherry fluorescence, the time gating method was applied with a gating time at 0.5–12 ns to eliminate background autofluorescence[68]. The excitation and emission wavelengths were 485 nm and 490–550 nm for GFP, 595 nm and 600–640 nm for mCherry, and 485 nm and 620–685 nm for chlorophyll fluorescence. The GFP signal at the outer edge of the guard cell was defined as the plasma membrane, whereas the signal inside the guard cell was defined as the cytosol. Fluorescence intensities in these regions were quantified using ImageJ 1.53q software (National Institutes of Health), and the cytosol/plasma membrane fluorescence intensity ratio was then calculated.

## BiFC assay

The BiFC assay was performed according to a previously described method, with slight modifications[27]. The *WDR48* cDNA sequence was amplified using primers 5′-TAGTGGATCCGTCGACATGCACCGGGTTGGAAGTGC-3′ and 5′-TACCCTCGAGGTCGACTTATCTAGCTATAGCCACTC-3′, and the product was inserted downstream of the SPYNE(R) 173 sequence at the *Sal* I site. *PHOT1* and *PIN1* cDNA sequences were amplified using the following primers: 5′-TAGTGGATCCGTCGACATGGAACCAACAGAAAAACCATCGAC-3′ and 5′-TACCCTCGAGGTCGACTCAAAAAACATTTGTTTGCAGATCTTC-3′ for *PHOT1*, and 5′-TAGTGGATCCGTCGACATGATTACGGCGGCGGACTTC-3′ and 5′-TACCCTCGAGGTCGACTCATAGACCCAAGAGAATGTAG-3′ for *PIN1*. The resulting products were inserted downstream of the SPYCE (MR) sequence using the *Sal* I site. To construct the *p35S:KAT1-mCherry* vector, the *mCherry* sequence was inserted into the *Sal* I/*Bam*H I sites of pPCS201, generating pPCS205. The genomic sequence of *KAT1* was amplified using primers 5′-ACATATGCCCGTCGACATGTCGATCTCTTGGACTCGAAATTTC-3′ and 5′-TGCTCACCATGTCGACATTTGATGAAAAATACAAATGATCACCATC-3′, and the product was inserted into the *Sal* I site of pPCS205. To construct the *p35S:OEP7-mCherry* vector, the cDNA sequences of *OEP7* and *mCherry* were amplified using the following primers: 5′-ACATATGCCCGTCGACATGGGAAAAACTTCGGGAGCG-3′ and 5′-GCCCTTGCTCACCATGTCTTTGGTTGGGTCAGATTTG-3′ for *OEP7*, and 5′-ATGGTGAGCAAGGGCGAGGAG-3′ and 5′-TTCAGAATTCGGATCCTTACTTGTACAGCTCGTCCATG-3′ for *mCherry*. The products were inserted into the *Sal* I/*Bam*H I sites of pRI 101-AN. To construct the *p35S:3×FLAG-WDR48* vector, the *3×FLAG* sequence was inserted into the *Sal* I/*Bam*H I sites of pRI 101-AN to yield pPCS106. The *WDR48* cDNA sequence was amplified using primers 5′-CGATGACAAGGGGATCCATGCACCGGGTTGGAAGTGC-3′ and 5′-TTCAGAATTCGGATCCTTATCTAGCTATAGCCACTCTGTAGTTG-3′, and the product was inserted into the *Bam*H I site of pPCS106.

The resulting vectors were transformed into mesophyll cell protoplasts using PEG-calcium transfection[69]. The adaxial epidermis of leaves from 4-week-old plants was removed, and the leaves were incubated in an enzyme solution containing 20 mM MES-KOH (pH 5.4), 0.25 M mannitol, 1% (w/v) cellulase R-10, 0.25% (w/v) macerozyme R-10, 0.1% (w/v) bovine serum albumin, 10 mM CaCl$_2$, and 20 mM KCl for 1 h. Mesophyll cell protoplasts were filtered through a 58 μm nylon mesh

and collected by centrifugation at $100 \times g$ for 3 min. The protoplasts were resuspended in W5 solution containing 2 mM MES-KOH (pH 5.7), 5 mM glucose, 154 mM NaCl, 125 mM CaCl$_2$, and 5 mM KCl and washed twice with W5 solution. The protoplasts were then resuspended in a buffer containing 4 mM MES-KOH (pH 5.7), 0.4 M mannitol, and 15 mM MgCl$_2$ and mixed with 5 μg of each vector and 25 μg of competitor vector. Transformation was initiated by adding an equivalent volume of PEG solution containing 40% (w/v) polyethylene glycol 4000, 0.2 M mannitol, and 100 mM CaCl$_2$. The reaction was halted by adding W5 solution, followed by centrifugation at $1,000 \times g$ for 1 min. The protoplasts were resuspended in W5 solution and incubated for 24 h in the dark. YFP and mCherry fluorescence were detected using a confocal laser scanning microscope (TCS SP8X; Leica Microsystems) with a time-gating window of 0.5–12 ns to eliminate background autofluorescence[68]. The excitation and emission wavelengths were 514 nm and 520–550 nm for YFP, and 595 nm and 600–640 nm for mCherry. YFP and mCherry signals were quantified using ImageJ 1.53q software (National Institutes of Health), and the relative YFP fluorescence intensity was calculated based on the mCherry signal.

## Chloroplast movements

Chloroplast movements were examined using green and white band assays, following previously described methods with slight modifications[39]. Leaves from dark-adapted plants were placed on 1% (w/v) agar plate, and slotted aluminum sheets were placed on top of the leaves. For the green band assay, leaves were illuminated with blue light (0.5 μmol m$^{-2}$ s$^{-1}$) for 1 h. For the white band assay, leaves were illuminated with blue light (50 μmol m$^{-2}$ s$^{-1}$) for 1 h.

## Measurement of lateral root number and root length

Plants were grown for 11 days on 0.8% (w/v) agar plates containing 1× Murashige–Skoog (MS) salts, 25 mM MES-NaOH (pH 5.7), and 0.5% (w/v) sucrose under continuous white light (50 μmol m$^{-2}$ s$^{-1}$) at 23 °C, as described previously[36]. Total lateral root number and root length were measured using ImageJ 1.53q software (National Institutes of Health).

## Statistical analysis

Reported data are presented as mean ± the standard error (SEM). Significant differences were determined by analysis of variance (ANOVA) with Tukey's test or Student's *t*-test using Microsoft Excel and Excel Toukei ver. 6.05 and 8.0 (Esumi).

## Reporting summary

Further information on research design is available in the Nature Portfolio Reporting Summary linked to this article.

## Data availability

The MS proteomics data have been deposited in the ProteomeXchange database under accession code PXD006586, PXD039740, and PXD073138 via jPOSTrepo[70]. Source data are provided with this paper.

## Code availability

The custom code for starch granule area quantification and the trained model used in this study have been deposited in GitHub (code: https://github.com/ShotaYamauchi/Starch-area-quantification; trained model: https://github.com/ShotaYamauchi/Starch-area-quantification/releases). A stable release containing both the code and the trained model has been archived on Zenodo (https://doi.org/10.5281/zenodo.18689938). This program is designed to be executed in the Google Colaboratory environment.

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

## Acknowledgements

We thank Asami Hiyama and Mika Machiki for assistance with phosphoproteomic analysis, Minoru Noguchi and Shintaro Ichikawa for assistance with confocal microscopy, and Ryuichi Nishihama, Kazuyuki Kuchitsu, and Michito Tsuyama for the equipment support. We also thank the Nottingham Arabidopsis Stock Centre (NASC) for providing the seeds used in this study. This work was supported by JSPS KAKENHI (grant number 21H02511 and 24K02045 to A.T., grant number 19K16171 and 22K15144 to S.Y., and grant number 21H02466 to N.S.), MEXT KAKENHI (grant number 21H05665, 22H04726, 23H04202, and 25H01347 to A.T., grant number 20H05906 to Y.T., and grant number 22H4735 to K.H.), JST ASPIRE (grant number JPMJAP24A1 to A.T.), the Japan Foundation for Applied Enzymology to A.T., Takeda Science Foundation to A.T., Yamaguchi University Project for Formation of the Core Research Project to A.T., Yamaguchi University Fund for Young Scientist Research to S.Y., and Swiss National Science Foundation (SNF grant 310030_185241/1) to D.S.

## Author contributions

S.Y. and A.T. conceived and designed the research; S.Y., S.F., H.I., N.S., Y.K., L.D., H.F., K.Y. and H.T. performed the experiments; S.Y., N.S., Y.K., T.U., K.H., D.S., K.S., and A.T. analysed the data; M.N. and Y.T. provided the in vitro transcription/translation system; and S.Y., N.S., K.H., and A.T. wrote the manuscript with input from all authors.

## Competing interests

The authors declare no competing interests.
