## [Transparent Peer Review file · Nature Communications]

Phosphorylation of WDR48 by phototropins drives starch degradation to promote stomatal opening

Corresponding Author: Professor Atsushi Takemiya

Version 0:

Reviewer comments:

Reviewer #1

(Remarks to the Author)

In this manuscript, the authors investigate the novel function of WDR48 in the blue light-mediated guard cell starch degradation and stomatal opening. Through phosphoproteome analysis, the authors identified WDR48 as a protein that is phosphorylated when exposed to blue light. Furthermore, PHOT1 was found to interact with and phosphorylate WDR48 at Ser 393. WDR48 and BLUS1 regulate guard cell starch degradation and stomatal opening through distinct pathways. The data presented in the manuscript is original and will be of interest to a wide range of researchers studying light signaling and stomatal movement. Some of the findings could be further supported by addressing the following concerns:

1. The authors conducted the phosphoproteomic analysis using guard cell protoplasts from wild type and phot1-5 phot2-1 mutants. However, it is unclear which materials were used in Figure 1a and 1b, and this information should be included in the figure legend. Furthermore, the manuscript does not indicate how many times these experiments were repeated and it will be better to perform the statistical analysis.
2. Blue light induces the phosphorylation of WDR48 via PHOT1/PHOT2, which promotes starch degradation in guard cells. It is unclear whether mimicking the phosphorylation of WDR48-S393E could alleviate the defective phenotype of impaired guard cell starch degradation and stomatal opening observed in phot1 phot2 mutants. In other words, as a downstream component of the PHOT1/PHOT2-mediated blue light signaling pathway, activated WDR48 should repress the defective phenotype of phot1phot2.
3. What is the biological significance of PHOT1-dependent phosphorylation modification of WDR48 protein, and how does this affect the protein localization of WDR48-S393A or D? Does this phosphorylation modification influence the binding ability of WDR48 to deubiquitinating enzymes (DUBs)?
4. It has been reported that WDR48 regulates lateral root initiation. It is unclear whether the blue light-induced phosphorylation of WDR48 has any effect on the regulation of lateral root growth by WDR48.
5. In Figure 5a, the confocal image depicts PHOT1-GFP localized in the plasma membrane, whereas GFP-WDR48 is found both in the plasma membrane and cytoplasm. It would be beneficial to perform a statistical analysis to determine whether blue light specifically regulates the subcellular localization of GFP-WDR48.
6. The BiFC experiments indicate that both PHOT1 and WDR48 are localized in the chloroplast, and PHOT1/PHOT2 play critical roles in regulating chloroplast movement in response to light. It is unclear whether WDR48 is also involved in this process. Additionally, it would be helpful if Figure 5a included an image depicting the chloroplast localization of PHOT1 and WDR48.
7. The observation of blue light-induced phosphorylation of WDR48 in the *aha1-9* mutant suggests that H⁺-ATPase is not required for this process. However, this data does not support the hypothesis that WDR48 regulates guard cell starch degradation through an H⁺-ATPase-independent pathway. It would be interesting to investigate whether activated WDR48-S393E can alleviate the defective starch degradation phenotype observed in *aha1-9*, *blus1*, and *cbc1* mutants.
8. All gel images should include the molecular weight marker and original image, and the relative quantified protein band

intensity compared to the non-phosphorylated protein should be presented.

Reviewer #2

(Remarks to the Author)

The current manuscript aims to explore the contributions of WDR48 in mediating the starch degradation and stomatal opening via the phosphorylation of blue light irradiated Phototropins. The authors identified WDR48 as a phosphorylated protein using mass spectrometry analysis under blue light conditions. By employing a mix of biochemical assays, PS-PI staining, confocal imaging, and immunoprecipitation, the authors propose that the phosphorylation of WDR48 is associated with blue light-induced starch degradation and stomatal opening. Additionally, this study establishes that WDR48 is phosphorylated by Phototropins. This work also presents other interesting claims, including the separate signaling pathways of WDR48 and BLUS1, which coordinately regulate starch degradation and synthesis. However, weaknesses in the data must be addressed before these claims can be fully supported.

There are several major limitations to this work:

1- The majority of data regarding the phosphorylation of WDR48 was obtained solely through the use of phospho-specific antibodies against Ser-393 of WDR-48. Considering the phosphorylation of WDR48 is a central claim in this paper and the authors have access to the mass spec facilities, presenting mass spectrometry data to support the key phosphorylation results presented via Western blot would be highly beneficial.

Secondly, based on mass spec data (Fig. 1b), the phosphorylation of WDR48 in blue light is noticeably stronger than that in red light (the peak of phosphopeptide containing pSer-393 of WDR48 is almost undetectable in RL compare to that of in BL). However, in most of WB assay (e.g. Fig. 1d), the band of WDR48 is clearly presented in RL of WT and in phot1-5/phot2-1. The authors should address the discrepancy between the mass spec and WB data, or consider the possibility that the antibody targeting Ser-393 may detect a non-specific protein with similar electrophoretic mobility to phosphorylated WDR48 (if this is the case, additional mass spec data for WDR48 phosphorylation should be presented).

2-The resolution of all the figures should be significantly improved. The current quality of figures on this MS does not meet the standards expected for publication in Nature Communications IMO. Most starch granules stained with PS-PI are hard to distinguish in this MS, especially those undergoing degradation. The authors could refer to ref.#28 to modify their figure (the current biology paper: "Blue light induces a distinct starch degradation pathway in guard cells for stomatal opening.").

3-To examine whether WDR48 interacts with phototropins, the authors detected the interaction of WDR48 and phototropins by in vitro pull down, co-immunoprecipitation and BiFC assay. However, the input data of the bait were missing in the pull down (Fig. 4f) and co-immunoprecipitation (Fig. 4g) assay. To me, the similar amount of prey (input of bait) is also important to assess the protein-protein interaction.

The Lines 202—204: The authors mention that "In the dark, GFP fluorescence of PHOT1-GFP was detected at the plasma membrane, and partial GFP signals were detected in the cytoplasm after blue light irradiation (Fig. 5a)." Regarding the BiFC assay, the authors should specify the light conditions under which it was conducted, considering that no PHOT1-GFP signal was detected in the chloroplast envelope in the dark.

Since the interaction and phosphorylation of WDR48 are the main claims of this paper, the authors should illustrate whether WDR48 interacts with and is phosphorylated by phototropins in dark via pull down, co-immunoprecipitation and in vitro kinase assay, which could support the phosphorylation data detected by antibodies against Ser-393 of WDR-48.

Other comments:

(1) The Confocal images of PS-PI staining (Fig. 2e) do not match the Quantification data (Fig. 2f). e.g. the starch granule of phot1-5/phot2-1 at 5 and 10 min.

(2) The trend of changes in stomatal aperture (Fig. 2g) is inconsistent with the curve in Fig. 2e. A confocal image depicting a consistent trend can be acquired to complement Fig. 2g.

(3) #3 line of GFP-S393A exhibits a discernible disparity from #9. Did the authors examine the protein expression levels of both lines?

(4) The Confocal images of Fig. 3a do not correspond to the statistical findings (both starch granule area and stomatal aperture) reported by the authors. Alternatively, it is possible that my interpretation was influenced by the poor quality of the images.

(5) In Figure 4, it can be observed that BLUS1 protein (using anti-BLUS1) exhibits distinct shifted bands indicative of phosphorylation after blue light irradiation. However, such shifted bands are not visible in the case of WDR48 (using anti-WDR48), and the phosphorylation of WDR48 can only be detected using anti-pSer393. This is one of my concerns regarding the phosphorylation detection in this study.

(6) The phosphorylation of WDR48 in the in vitro kinase assay (Fig. 4e) should be confirmed by λ -PPase.

(7) The author should explain that, in Fig. 4b, why the band related to anti-pSer393 was not detectable in WT under RL, compared to the same situation in other figures, e.g. Fig. 4a.

(8) Since phot1-GFP refers to a protein, it should be written as PHOT1-GFP.

Version 1:

Reviewer comments:

Reviewer #1

(Remarks to the Author)

In the revised manuscript, the authors have effectively addressed most of my concerns. The paper is now more comprehensive and supported by stronger data. I have a few minor suggestions to further enhance the quality of this manuscript.

1. It would be beneficial to present a new working model that including more information.
2. The images in Figure 5A seem inconsistent with those in Supplementary Figures 8 and 9, particularly regarding the GFP-WDR48 images. In Figure 5A, GFP-WDR48 appears relatively blurry and seems to predominantly localize to the cytoplasm, whereas in Supplementary Figures 8 and 9, GFP-WDR48 displays clearer localization at the plasma membrane and chloroplast envelope, similar to GFP-PHOT1.
3. There are several spelling errors throughout the manuscript. For instance, "recognise" in line 118 should be corrected to "recognize," and "localisation" in line 226 should be changed to "localization."

Reviewer #2

(Remarks to the Author)

I consider my comments to have been fully addressed, and I have no further questions.

RESPONSE TO REVIEWER COMMENTS

Reviewer #1 (Remarks to the Author):

In this manuscript, the authors investigate the novel function of WDR48 in the blue light-mediated guard cell starch degradation and stomatal opening. Through phosphoproteome analysis, the authors identified WDR48 as a protein that is phosphorylated when exposed to blue light. Furthermore, PHOT1 was found to interact with and phosphorylate WDR48 at Ser 393. WDR48 and BLUS1 regulate guard cell starch degradation and stomatal opening through distinct pathways. The data presented in the manuscript is original and will be of interest to a wide range of researchers studying light signaling and stomatal movement. Some of the findings could be further supported by addressing the following concerns:

1. The authors conducted the phosphoproteomic analysis using guard cell protoplasts from wild type and *phot1-5 phot2-1* mutants. However, it is unclear which materials were used in Figure 1a and 1b, and this information should be included in the figure legend. Furthermore, the manuscript does not indicate how many times these experiments were repeated and it will be better to perform the statistical analysis.

Response: Thank you for your suggestions. We have added the information of the materials used for the phosphoproteome analysis to the figure legend of Fig. 1a and 1b. We also provided the relative peak area of the phosphopeptide containing pSer-393 of WDR48 in the wild type and the *phot1 phot2* mutant, along with the number of replicates and the results of statistical analysis.

2. Blue light induces the phosphorylation of WDR48 via PHOT1/PHOT2, which promotes starch degradation in guard cells. It is unclear whether mimicking the phosphorylation of WDR48-S393E could alleviate the defective phenotype of impaired guard cell starch degradation and stomatal opening observed in *phot1 phot2* mutants. In other words, as a downstream component of the PHOT1/PHOT2-mediated blue light signaling pathway, activated WDR48 should repress the defective phenotype of *phot1phot2*.

Response: Thank you for your insightful comment. To address this point, we crossed the transgenic plants expressing GFP-S393E (generated in the *wdr48-1* mutant background) with the *phot1-5 phot2-1* double mutant, thereby obtaining plants expressing GFP-S393E in the *phot1-5 phot2-1 wdr48-1* triple-mutant background. We then examined starch degradation and stomatal opening in response to blue light and fusicoccin, a plasma membrane H⁺-ATPase activator (Fig. 6g-i).

In these plants, we found that blue light-induced starch degradation and stomatal opening were not restored, indicating that the phosphomimetic S393E mutation alone is insufficient to rescue these defects in the absence of *phot1* and *phot2*. In contrast, fusicoccin treatment successfully induced starch degradation and stomatal opening, even without blue light irradiation.

These observations suggest the following interpretation. Although the S393E substitution likely renders WDR48 constitutively active in a manner similar to its phosphorylated state, the loss of *phot1* and *phot2* prevents blue light-dependent phosphorylation of BLUS1, a key upstream regulator required for activation of the plasma membrane H⁺-ATPase. As a result, the plasma membrane hyperpolarisation and the downstream starch degradation pathway, which depends on H⁺-ATPase activity, are all impaired in the *phot1-5 phot2-1* background. This explains why blue light-induced starch degradation and stomatal opening does not occur even when WDR48 is constitutively activated.

In contrast, fusicoccin bypasses the phototropin–BLUS1 pathway by directly activating H⁺-ATPase, thereby restoring plasma membrane hyperpolarisation and initiating the downstream signalling that leads to starch degradation. Additionally, because WDR48 is already activated by the S393E mutation, both signalling branches are functional under fusicoccin treatment, resulting in the observed induction of starch degradation and stomatal opening.

Taken together, these findings indicate that although the phosphorylation-mimicking activation of WDR48 is important, it is not sufficient to compensate for the loss of the phototropin–BLUS1–H⁺-ATPase signalling cascade, which is essential for blue light-induced guard cell responses.

3. What is the biological significance of PHOT1-dependent phosphorylation modification of WDR48 protein, and how does this affect the protein localization of WDR48-S393A or D? Does this phosphorylation modification influence the binding ability of WDR48 to deubiquitinating enzymes (DUBs)?

Response: Thank you for your insightful questions. At present, the biological significance of phototropin-mediated phosphorylation of WDR48 remains unclear. We have conducted additional analyses on the subcellular localisation of GFP-S393A and GFP-S393E, but no noticeable differences were observed compared with GFP-WDR48, indicating that this phosphorylation modification does not affect WDR48 localisation (Supplementary Fig. 8).

Regarding the potential influence of WDR48 phosphorylation on its interaction with deubiquitinating enzymes (DUBs), we acknowledge that such an effect is conceivable. However, investigating this question is beyond the scope of the current study, as it is not yet established whether WDR48 interacts with DUBs in guard cells or whether DUBs contribute to starch degradation and stomatal opening. We consider this an interesting topic for future research.

4. It has been reported that WDR48 regulates lateral root initiation. It is unclear whether the blue light-induced phosphorylation of WDR48 has any effect on the regulation of lateral root growth by WDR48.

Response: Thank you for your insightful comment. We indeed confirmed not only the inhibition of lateral root formation but also the suppression of primary root growth in the *wdr48* mutants (Supplementary Fig. 5). However, these phenotypes were complemented by the introduction of wild-type WDR48 as well as by the phosphodeficient and phosphomimetic forms of WDR48. Thus, it appears that these root developmental processes require functions of WDR48 that are independent of its phosphorylation status. We have incorporated these findings into the Results section.

5. In Figure 5a, the confocal image depicts PHOT1-GFP localized in the plasma membrane, whereas GFP-WDR48 is found both in the plasma membrane and cytoplasm. It would be beneficial to perform a statistical analysis to determine whether blue light specifically regulates the subcellular localization of GFP-WDR48.

Response: Thank you for this valuable suggestion. In response to your comment, we performed a quantitative analysis of fluorescence intensity in both the plasma membrane and cytosolic fractions, and evaluated their relative distribution using the cytosol/plasma membrane ratio for GFP-WDR48 as well as for phot1-GFP (Fig. 5a and 5b).

Our statistical analysis revealed that phot1-GFP exhibited a significant increase in the cytosol/plasma membrane ratio upon blue light irradiation, consistent with the partial release of phot 1 from the plasma membrane into the cytosol under blue light conditions (Sakamoto and Briggs, 2002; Wan et al., 2008). In contrast, GFP-WDR48 did not show any detectable change in the cytosol/plasma membrane ratio following blue light exposure, indicating that its subcellular localisation remains stable during the blue light response.

These results demonstrate that, unlike phot1, the localisation of WDR48 is not dynamically regulated by blue light within the timescale examined. We have incorporated these new quantitative data into the revised manuscript (Fig. 5b).

(Reference)

Sakamoto and Briggs (2002) *Plant Cell* 14: 1723–1735.

Wan et al. (2008) *Mol Plant* 1: 103–117.

6. The BiFC experiments indicate that both PHOT1 and WDR48 are localized in the chloroplast, and PHOT1/PHOT2 play critical roles in regulating chloroplast movement in response to light. It is unclear whether WDR48 is also involved in this process. Additionally, it would be helpful if Figure 5a included

an image depicting the chloroplast localization of PHOT1 and WDR48.

Response: We appreciate the reviewer's insightful comments regarding the potential involvement of WDR48 in chloroplast movements and the localisation of phot1 and WDR48 on the chloroplast envelope. To address these points, we performed additional analyses using the *wdr48* mutants. Our results demonstrate that both the weak blue light-induced chloroplast accumulation and the strong blue light-induced chloroplast avoidance occur normally in the *wdr48* mutants, comparable to those in the wild type (Supplementary Fig. 3). These findings indicate that WDR48 is not essential for chloroplast movements.

In addition, following the reviewer's suggestion regarding Figure 5a, we conducted additional analyses to examine the chloroplast envelope localisation of phot1-GFP and GFP-WDR48 in more detail. We introduced the chloroplast envelope marker OEP7-mCherry into plants expressing phot1-GFP or GFP-WDR48 and confirmed that the GFP signals of both fusion proteins colocalised with OEP7-mCherry under both dark and blue light conditions. These findings indicate that both phot1 and WDR48 are partially localised to the chloroplast envelope irrespective of light conditions.

We have included representative images illustrating their colocalisation on the chloroplast envelope, as well as images depicting the chloroplast region, which now provided as Supplementary Figs. 9a and 9b, respectively.

7. The observation of blue light-induced phosphorylation of WDR48 in the *aha1-9* mutant suggests that H⁺-ATPase is not required for this process. However, this data does not support the hypothesis that WDR48 regulates guard cell starch degradation through an H⁺-ATPase-independent pathway. It would be interesting to investigate whether activated WDR48-S393E can alleviate the defective starch degradation phenotype observed in *aha1-9*, *blus1*, and *cbc1* mutants.

Response: Thank you for this insightful comment. Following our analysis in the *phot1-5 phot2-1* background described above, we conducted similar experiments to evaluate whether activated WDR48-S393E can alleviate the defects in guard cell starch degradation observed in *blus1-1* and *aha1-9* mutants.

To address this, we crossed the *wdr48-1* expressing GFP-S393E with *blus1-1* and *aha1-9* mutants, and obtained plants expressing GFP-S393E in the *blus1-1 wdr48-1* and *aha1-9 wdr48-1* mutant backgrounds. We then examined starch degradation and stomatal opening in response to blue light and fusicoccin (Fig. 6g–i).

In the *blus1-1 wdr48-1* background expressing GFP-S393E, blue light did not induce starch degradation or stomatal opening, consistent with the results in the *phot1-5 phot2-1 wdr48-1* background described above. However, fusicoccin treatment successfully induced both starch

degradation and stomatal opening, indicating that although S393E is constitutively active, BLUS1-dependent H⁺-ATPase activation remains essential for the blue light-triggered response.

In contrast, in the *aha1-9 wdr48-1* background expressing GFP-S393E, neither blue light nor fusicoccin triggered starch degradation or stomatal opening. Since the *aha1-9* mutant exhibits reduced expression of the H⁺-ATPase, it is conceivable that even fusicoccin is unable to sufficiently induce plasma membrane hyperpolarisation and downstream signalling required for starch degradation. This explains why GFP-S393E cannot compensate for the defective phenotype in this background.

Although the data are not shown, blue light-dependent starch degradation in the *cbc1-1 cbc2-1* double mutant occurred normally, similar to the wild type. Therefore, we did not conduct further analyses using plants expressing GFP-S393E in the *cbc1-1 cbc2-1 wdr48-1* mutant background.

Taken together, these results collectively suggest that although the phosphomimetic S393E mutation activates WDR48, its ability to promote starch degradation still depends on a functional level of H⁺-ATPase activity. Thus, WDR48 alone is insufficient to drive starch degradation when the H⁺-ATPase activation is severely compromised, as in the *blus1-1* and *aha1-9* mutants. The results described above, including those obtained using GFP-S393E in the *phot1 phot2 wdr48* mutant background, have been incorporated into the revised manuscript, and the implications of these findings are thoroughly discussed in the Discussion section.

8. All gel images should include the molecular weight marker and original image and the relative quantified protein band intensity compared to the non-phosphorylated protein should be presented.

Response: Thank you for your comments. We have included the molecular weight information in all gel images (Figs. 1d, 1g, 1k, 1m, 1n, 4a, 4c, 4e, 4g, 4i, 4k, and 4m; Supplementary Fig. 1a, 1c, 1e, 4a; 7a, 7c, 7e, 10a, and 10c), and we have provided the original gel images in the Source Data. In addition, we have provided the quantitative data for the relative phosphorylation levels (Figs. 1e, 1i, 4b, 4d, 4f, and 4h; Supplementary Fig. 1b, 1d, 1f, 7b, 7d, 7f, 10b, and 10d).

Reviewer #2 (Remarks to the Author):

The current manuscript aims to explore the contributions of WDR48 in mediating the starch degradation and stomatal opening via the phosphorylation of blue light irradiated Phototropins. The authors identified WDR48 as a phosphorylated protein using mass spectrometry analysis under blue light conditions. By employing a mix of biochemical assays, PS-PI staining, confocal imaging, and immunoprecipitation, the authors propose that the phosphorylation of WDR48 is associated with blue light-induced starch degradation and stomatal opening. Additionally, this study establishes that

WDR48 is phosphorylated by Phototropins. This work also presents other interesting claims, including the separate signaling pathways of WDR48 and BLUS1, which coordinately regulate starch degradation and synthesis. However, weaknesses in the data must be addressed before these claims can be fully supported.

There are several major limitations to this work:

1- The majority of data regarding the phosphorylation of WDR48 was obtained solely through the use of phospho-specific antibodies against Ser-393 of WDR-48. Considering the phosphorylation of WDR48 is a central claim in this paper and the authors have access to the mass spec facilities, presenting mass spectrometry data to support the key phosphorylation results presented via Western blot would be highly beneficial. Secondly, based on mass spec data (Fig. 1b), the phosphorylation of WDR48 in blue light is noticeably stronger than that in red light (the peak of phosphopeptide containing pSer-393 of WDR48 is almost undetectable in RL compare to that of in BL). However, in most of WB assay (e.g. Fig. 1d), the band of WDR48 is clearly presented in RL of WT and in *phot1-5/phot2-1*. The authors should address the discrepancy between the mass spec and WB data, or consider the possibility that the antibody targeting Ser-393 may detect a non-specific protein with similar electrophoretic mobility to phosphorylated WDR48 (if this is the case, additional mass spec data for WDR48 phosphorylation should be presented).

Response: We thank the reviewer for the insightful comments regarding the phosphorylation of WDR48. Initially, we performed phosphoproteomic analysis using guard cell protoplasts isolated from Arabidopsis wild type, and identified WDR48 as a blue light-dependent phosphorylated protein. As guard cell protoplasts can only be obtained in small amounts from a large quantity of plant leaves, it is not feasible to perform phosphoproteomic analysis for all biochemical experiments. Therefore, we generated phospho-specific antibodies recognizing WDR48 phosphorylated at Ser-393 and used them to assess WDR48 phosphorylation under various conditions and in different mutant backgrounds.

To address the reviewer's concerns, we reperformed phosphoproteomic analysis using guard cell protoplasts from both wild type and *phot1-5 phot2-1* mutant. Quantitative analysis of the Ser-393-containing phosphopeptide revealed that this phosphopeptide is indeed detectable under red light in the wild type, and its abundance increases by approximately 3.5-fold upon blue light irradiation. In the *phot1-5 phot2-1* mutant, the phosphopeptide is detected under red light at a level similar to that of wild-type red light samples; however, blue light irradiation does not increase its abundance. These quantitative values have been added to Fig. 1b.

We also conducted a detailed validation of the specificity of the phospho-specific antibodies against Ser-393 of WDR48 used in our biochemical analyses. In wild-type guard cell protoplasts, a weak signal was detectable even under red light, and the signal increased upon blue light irradiation. λ -

protein phosphatase treatment reduced the signal to below the detection limit under both red and blue light conditions (Supplementary Figs. 1a and 1b). In the *wdr48* null mutant, no signal was detected at the molecular weight corresponding to WDR48, ruling out cross-reactivity with other proteins of similar size (Supplementary Figs. 1c and 1d). Furthermore, in the phosphodeficient WDR48 mutant (S393A), the antibodies did not detect any band at the WDR48 molecular weight (Supplementary Figs. 1e and 1f). These results confirm that the antibodies specifically recognise Ser-393-phosphorylated WDR48.

Finally, we performed quantitative analysis for all biochemical assays using these antibodies (Fig. 1e, 4b, 4d, 4f, and 4h; Supplementary Figs. 1b, 1d, 1f, 7b, 7d, 7f, 10b, and 10d). As an example, in wild type and the *phot1-5 phot2-1* mutant, the relative phosphorylation level (with wild-type blue light set as 100%) showed approximately 20% signal under red light for both genotypes, and no increase was observed in *phot1-5 phot2-1* plants under blue light (Fig. 1e). These results are consistent with the quantitative data from phosphoproteomic analysis (Fig. 1b).

Taken together, these results demonstrate that WDR48 undergoes low-level basal phosphorylation under red light in guard cells, that its phosphorylation levels are strongly enhanced by blue light, and that this blue light-dependent phosphorylation is dependent on phototropins.

2-The resolution of all the figures should be significantly improved. The current quality of figures on this MS does not meet the standards expected for publication in Nature Communications IMO. Most starch granules stained with PS-PI are hard to distinguish in this MS, especially those undergoing degradation. The authors could refer to ref.#28 to modify their figure (the current biology paper: "Blue light induces a distinct starch degradation pathway in guard cells for stomatal opening.").

Response: Thank you for your valuable comment. We have carefully reexamined the starch granule data and found that some of the original images did not clearly represent the degradation status of starch granules in guard cells. To address this, we have replaced the images with higher resolution confocal images that more clearly depict starch granule morphology and degradation. We believe these updated images improve the visual clarity and accurately correspond to the quantified data, ensuring that the figures meet the standards expected for publication.

3-To examine whether WDR48 interacts with phototropins, the authors detected the interaction of WDR48 and phototropins by in vitro pull down, co-immunoprecipitation and BiFC assay. However, the input data of the bait were missing in the pull down (Fig. 4f) and co-immunoprecipitation (Fig. 4g) assay. To me, the similar amount of prey (input of bait) is also important to assess the protein-protein interaction.

Response: Thank you for your comment. In response, we have repeated the *in vitro* pull-down and co-immunoprecipitation assays and included the input data for both the prey and the bait (Figs. 4i, 4k, and 4m). The reanalysed results confirm the interaction between phototropins and WDR48, consistent with our original findings.

The Lines 202—204: The authors mention that “In the dark, GFP fluorescence of PHOT1-GFP was detected at the plasma membrane, and partial GFP signals were detected in the cytoplasm after blue light irradiation (Fig. 5a).” Regarding the BiFC assay, the authors should specify the light conditions under which it was conducted, considering that no PHOT1-GFP signal was detected in the chloroplast envelope in the dark.

Response: We appreciate the reviewer’s insightful comment regarding the light conditions used for the BiFC assay. The BiFC experiments were performed using dark-treated protoplasts. During our observations, we noted that in the dark, phot1-GFP fluorescence is mainly observed at the plasma membrane, with weaker signals occasionally detectable around chloroplasts, and that blue light irradiation enhances its cytoplasmic localisation, as shown in Fig. 5a.

To clarify its localisation on the chloroplast envelope more precisely, we introduced the chloroplast envelope marker OEP7-mCherry into plants expressing phot1-GFP, as well as those expressing GFP-WDR48, and performed additional analyses (Supplementary Fig. 9). These analyses revealed that phot1-GFP signals are detectable on the chloroplast envelope under both dark and blue light conditions, consistent with a previous report (Kong et al., 2013), which was confirmed by colocalisation with OEP7-mCherry. Similarly, GFP-WDR48 also localises to the chloroplast envelope in both conditions.

These findings explain why BiFC signals appear on the chloroplast envelope even in dark-treated protoplasts and address the reviewer’s concerns regarding phot1-GFP localisation. We have revised the relevant section of the manuscript accordingly and incorporated these new data to more accurately reflect phot1-GFP and GFP-WDR48 behavior under dark and blue light conditions.

(Reference)

Kong et al. (2013) *Plant Cell Physiol* 54: 80–92.

Since the interaction and phosphorylation of WDR48 are the main claims of this paper, the authors should illustrate whether WDR48 interacts with and is phosphorylated by phototropins in dark via pull down, co-immunoprecipitation and *in vitro* kinase assay, which could support the phosphorylation data detected by antibodies against Ser-393 of WDR-48.

Response: Thank you for your comment. According to your comment, we reanalysed the interaction

between WDR48 and phototropins under both dark and blue light conditions using *in vitro* pull-down and co-immunoprecipitation assays. These analyses revealed that WDR48 binds to phototropins regardless of blue light exposure (Figs. 4i–l). Additionally, we performed *in vitro* kinase assays using full-length recombinant phototropins and confirmed the blue light-dependent phosphorylation of WDR48 (Figs. 4g and 4h; Supplementary Figs. 7a and 7b), consistent with *in vivo* phosphorylation data. These results further support our conclusion that WDR48 is phosphorylated by phototropins in a blue light-dependent manner.

Other comments:

(1) The Confocal images of PS-PI staining (Fig. 2e) do not match the Quantification data (Fig. 2f), e.g. the starch granule of *phot1-5/phot2-1* at 5 and 10 min.

Response: Thank you for your comment. We have replaced the images in Fig. 2e with the correct confocal images that accurately correspond to the quantification data shown in Fig. 2f. This ensures consistency between the visual representation of starch granules and the quantified measurements for all genotypes, including *phot1-5 phot2-1* at 5 and 10 min.

(2) The trend of changes in stomatal aperture (Fig. 2g) is inconsistent with the curve in Fig. 2e. A confocal image depicting a consistent trend can be acquired to complement Fig. 2g.

Response: Thank you for your comment. Indeed, in the data for starch degradation shown in Fig. 2e and 2f, starch in the wild type is largely degraded within 5 min of blue light irradiation. In contrast, stomatal opening (Fig. 2g) is initiated at 5 min but continues to increase until 10 min, showing a different time course.

This difference likely reflects the physiological process linking starch degradation to stomatal opening: as the degradation products must be converted into osmolytes such as glucose and malate, a process that requires time. Glucose functions as an osmotic regulator, while malate acts as a counterion for K^+ (Ogawa et al., 1978; Shimazaki et al., 2007; Dong et al., 2018; Flütsch et al. 2020). Together, these products drive water influx into guard cells, resulting in stomatal opening. Consequently, a temporal lag between the rapid starch degradation and the subsequent stomatal opening is expected. Therefore, the observed difference in time course between starch degradation and stomatal opening is consistent with the underlying physiological mechanism.

(Reference)

Ogawa et al. (1978) *Planta* 142: 61–65.

Shimazaki et al. (2007) *Annu Rev Plant Biol* 58: 219–247.

Dong et al. (2018) *Mol Plant* 11: 1278–1291.

Flütsch et al. (2020) *Plant Cell* 32: 2325–2344.

(3) #3 line of GFP-S393A exhibits a discernible disparity from #9. Did the authors examine the protein expression levels of both lines?

Response: Thank you for your comment. There is no statistically significant difference between the #3 and #9 lines of GFP-S393A in terms of blue light-dependent starch degradation and stomatal opening (Fig. 3a–c). We also examined the protein expression levels of both lines and found no statistically significant difference; both lines express GFP-S393A at levels comparable to the endogenous WDR48 in the wild type (Supplementary Figs. 4a and 4b).

(4) The Confocal images of Fig. 3a do not correspond to the statistical findings (both starch granule area and stomatal aperture) reported by the authors. Alternatively, it is possible that my interpretation was influenced by the poor quality of the images.

Response: Thank you for your comment. We have replaced the confocal images in Fig. 3a with higher-quality images that accurately correspond to the statistical findings for both starch granule area and stomatal aperture. This ensures that the visual representation is fully consistent with the quantified data and provides a clearer depiction of the observed phenotypes.

(5) In Figure 4, it can be observed that BLUS1 protein (using anti-BLUS1) exhibits distinct shifted bands indicative of phosphorylation after blue light irradiation. However, such shifted bands are not visible in the case of WDR48 (using anti-WDR48), and the phosphorylation of WDR48 can only be detected using anti-pSer393. This is one of my concerns regarding the phosphorylation detection in this study.

Response: Thank you for your insightful comment. While BLUS1 shows a clear band shift upon blue-light-dependent phosphorylation, WDR48 does not exhibit a detectable mobility shift in SDS-PAGE. Phosphorylation does not always result in a detectable mobility shift on SDS-PAGE. This difference likely reflects intrinsic properties of WDR48. Phosphorylation of WDR48 does not appear to induce a conformational change sufficient to alter its electrophoretic mobility, and the amino acid context around the phosphorylation site, including local charge distribution, may further prevent a visible shift.

Importantly, the phosphorylation of WDR48 under blue light has been independently confirmed

by phosphoproteomic analysis, in addition to detection using the phospho-specific antibodies (anti-pSer393) and λ -phosphatase treatment. These results collectively demonstrate that WDR48 is indeed phosphorylated under blue light, even though no mobility shift is observed on SDS-PAGE.

(6) The phosphorylation of WDR48 in the *in vitro* kinase assay (Fig. 4e) should be confirmed by λ -PPase.

Response: Thank you for your valuable comment. Following your suggestion, we performed an *in vitro* kinase assay using full-length phot1 and phot2 and confirmed the blue light-dependent phosphorylation of WDR48. We further verified that these phosphorylation signals were reduced by λ -PPase treatment. These results demonstrate that WDR48 is indeed phosphorylated by phot1 and phot2 in a blue light-dependent manner *in vitro*.

(7) The author should explain that, in Fig. 4b, why the band related to anti-pSer393 was not detectable in WT under RL, compared to the same situation in other figures, e.g. Fig. 4a.

Response: Thank you for pointing this out. In Fig. 4b, the anti-pSer393 signal in the wild type under RL was not detectable because the blot shown in the original version had been captured with a shorter exposure time when compared to other figures such as Fig. 4a. To avoid any misunderstanding, we have replaced the panel with a blot obtained using a longer exposure time, in which the band is clearly visible.

(8) Since phot1-GFP refers to a protein, it should be written as PHOT1-GFP.

Response: We understand your concern. According to the established nomenclature rules for phototropin proteins, the holoprotein containing the chromophore flavin mononucleotide (FMN) is denoted as phot, whereas the apoprotein lacking FMN is written as PHOT. Because phototropin translated *in vivo* predominantly exists as the holoprotein form, we have designated the protein as phot1-GFP throughout the manuscript.

(Reference)

Briggs et al. (2001) *Plant Cell* 13: 993–997.

RESPONSE TO REVIEWER COMMENTS

Reviewer #1 (Remarks to the Author):

In the revised manuscript, the authors have effectively addressed most of my concerns. The paper is now more comprehensive and supported by stronger data. I have a few minor suggestions to further enhance the quality of this manuscript.

1. It would be beneficial to present a new working model that including more information.

We thank the reviewer for the suggestion. In response, we have incorporated additional information into a new working model, presented in Fig. 7. This updated model includes starch synthesis via photosynthesis under red light and starch degradation mediated by phototropin-dependent phosphorylation of BLUS1 and WDR48 under blue light. It illustrates how red and blue light synergistically promote stomatal opening.

2. The images in Figure 5A seem inconsistent with those in Supplementary Figures 8 and 9, particularly regarding the GFP-WDR48 images. In Figure 5A, GFP-WDR48 appears relatively blurry and seems to predominantly localize to the cytoplasm, whereas in Supplementary Figures 8 and 9, GFP-WDR48 displays clearer localization at the plasma membrane and chloroplast envelope, similar to GFP-PHOT1.

We thank the reviewer for this careful observation. Upon reviewing all imaging data, we found that the confocal images of GFP-WDR48 shown in Fig. 5A did not capture the plasma membrane and chloroplast envelope signals well due to limitations in focal depth. We have now replaced these images with confocal images taken at similar focal depths to those in Supplementary Fig. 9 and 10, which more accurately represent the typical localisation pattern of GFP-WDR48. In addition, we have provided a series of confocal images of GFP-WDR48 acquired at different focal planes as Supplementary Fig. 8, to further demonstrate its subcellular localisation and to ensure transparency regarding the imaging conditions.

3. There are several spelling errors throughout the manuscript. For instance, "recognise" in line 118 should be corrected to "recognize," and "localisation" in line 226 should be changed to "localization."

We thank the reviewer for pointing this out. Since *Nature Communications* is published by a UK-based publisher, we have used British English throughout the manuscript. Therefore, spellings such as

“recognise” and “localisation” follow British English conventions.